# Towards Robust Multimodal Open-set Test-time Adaptation via Adaptive Entropy-aware Optimization

**Hao Dong**[1]  **Eleni Chatzi**[1]  **Olga Fink**[2]

[1]ETH Zürich  [2]EPFL
{hao.dong, chatzi}@ibk.baug.ethz.ch, olga.fink@epfl.ch

## Abstract

Test-time adaptation (TTA) has demonstrated significant potential in addressing distribution shifts between training and testing data. Open-set test-time adaptation (OSTTA) aims to adapt a source pre-trained model online to an unlabeled target domain that contains *unknown* classes. This task becomes more challenging when multiple modalities are involved. Existing methods have primarily focused on unimodal OSTTA, often filtering out low-confidence samples without addressing the complexities of multimodal data. In this work, we present Adaptive Entropy-aware Optimization (*AEO*), a novel framework specifically designed to tackle Multimodal Open-set Test-time Adaptation (MM-OSTTA) for the first time. Our analysis shows that the entropy difference between *known* and *unknown* samples in the target domain strongly correlates with MM-OSTTA performance. To leverage this, we propose two key components: Unknown-aware Adaptive Entropy Optimization (UAE) and Adaptive Modality Prediction Discrepancy Optimization (AMP). These components enhance the model's ability to distinguish unknown class samples during online adaptation by amplifying the entropy difference between known and unknown samples. To thoroughly evaluate our proposed methods in the MM-OSTTA setting, we establish a new benchmark derived from existing datasets. This benchmark includes two downstream tasks – action recognition and 3D semantic segmentation – and incorporates five modalities: video, audio, and optical flow for action recognition, as well as LiDAR and camera for 3D semantic segmentation. Extensive experiments across various domain shift scenarios demonstrate the efficacy and versatility of the *AEO* framework. Additionally, we highlight the strong performance of *AEO* in *long-term* and *continual* MM-OSTTA settings, both of which are challenging and highly relevant to real-world applications. This underscores *AEO*'s robustness and adaptability in dynamic environments. Our source code is available at https://github.com/donghao51/AEO.

## 1 Introduction

Test-time adaptation (TTA) significantly enhances the robustness and adaptability of machine learning models by enabling a source pre-trained model to adapt to target domains experiencing distribution shifts (Wang et al., 2021). This adaptability is crucial for ensuring the applicability of models in real-world scenarios, such as autonomous driving and action recognition. To address the challenges posed by distribution shifts, a variety of TTA algorithms have been developed (Niu et al., 2022; Yuan et al., 2023; Gong et al., 2024). These algorithms adapt specific model parameters using incoming test samples through unsupervised objectives such as entropy minimization and pseudo-labeling. However, most of these algorithms are designed for unimodal data, particularly images (Li et al., 2023). As real-world applications increasingly demand the processing of multimodal data, extending these approaches to support multimodal TTA across various modalities, including audio-video (Kazakos et al., 2019) and LiDAR-camera (Dong et al., 2022), has become essential. In response to this, several methods, such as READ (Yang et al., 2024) and MM-TTA (Shin et al., 2022), have been proposed to address the complexities inherent in multimodal TTA.

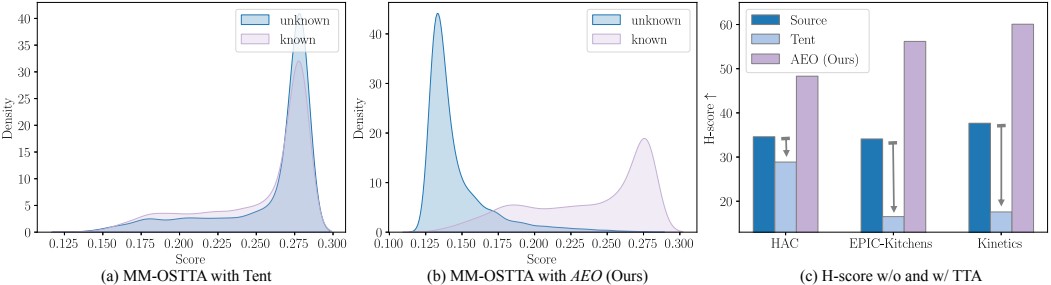

Figure 1: (a) Tent minimizes the entropy of all samples, making it difficult to separate the prediction score distributions of known and unknown samples. (b) Our *AEO* amplifies entropy differences between known and unknown samples through adaptive optimization. (c) As a result, Tent negatively impacts MM-OSTTA performance while *AEO* significantly improves unknown class detection.

A fundamental assumption in TTA is the alignment of label spaces between the source and target domains. However, real-world applications like autonomous driving (Blum et al., 2019) often involve target domains containing novel categories not present in the source label space (Nejjar et al., 2024). As a result, models adapted under this assumption may struggle with samples from these novel categories, significantly reducing the robustness of existing TTA methods (Fig. 1). This scenario, where the target domain contains *unknown* classes not present in the source domain, is referred to as open-set TTA. Several unimodal open-set TTA approaches, including OSTTA (Lee et al., 2023) and UniEnt (Gao et al., 2024), have been developed. However, OSTTA assumes that confidence values for unknown samples are lower in the adapted model than in the original model, which may not hold in Multimodal Open-Set TTA (MM-OSTTA) settings. UniEnt relies heavily on the quality of the embedding space to accurately detect unknown classes. The goal of MM-OSTTA is to adapt a pre-trained *multimodal model* from the source domain to a previously unseen target domain with the same modalities but including samples from *unknown* classes. The key challenge of MM-OSTTA is efficiently leveraging complementary information from diverse modalities to improve adaptation and unknown class detection – areas where current unimodal open-set TTA methods fall short.

Building on our observation that the entropy difference between known and unknown samples in the target domain is strongly correlated with the MM-OSTTA performance – where a larger entropy difference results in better detection of unknown classes – we introduce the novel Adaptive Entropy-aware Optimization (*AEO*) framework. *AEO* is designed to amplify the entropy difference between known and unknown samples during online adaptation and consists of two key modules: Unknown-aware Adaptive Entropy Optimization (UAE) and Adaptive Modality Prediction Discrepancy Optimization (AMP). UAE dynamically assigns weights to each sample based on an entropy threshold and automatically determines whether to minimize or maximize the entropy for each sample. AMP adjusts prediction discrepancies between different modalities adaptively. It encourages diverse predictions between modalities for unknown samples while maintaining consistent predictions for known samples. Together, these modules enable *AEO* to significantly amplify entropy differences between known and unknown samples during online adaptation, leading to substantial improvements in unknown class detection (Fig. 1 and Fig. 2).

To comprehensively evaluate the MM-OSTTA task, we develop a new benchmark derived from existing datasets. This benchmark includes two downstream tasks – *action recognition* and *3D semantic segmentation* – and incorporates five modalities: *video*, *audio*, and *optical flow* for action recognition, as well as *LiDAR* and *camera* for 3D semantic segmentation. Extensive experiments conducted across various domain shift scenarios demonstrate the efficacy and versatility of the proposed *AEO* framework. Furthermore, we evaluate *AEO* in challenging yet practical *long-term* and *continual* MM-OSTTA settings. *AEO* is robust against error accumulation and can constantly optimize the entropy difference between known and unknown samples over multiple rounds of adaptation, a capability essential for real-world dynamic applications. Our contributions can be summarized as follows:

- We explore the novel field of Multimodal Open-Set Test-time Adaptation, a concept with significant implications for real-world applications. MM-OSTTA involves adapting a pre-trained *multimodal model* from a source domain to a target domain that shares the same modalities but includes samples from previously *unknown* classes.

- To address MM-OSTTA, we propose Adaptive Entropy-aware Optimization, which effectively amplifies the entropy difference between known and unknown samples during online adaptation. Additionally, we establish a new benchmark based on existing datasets to comprehensively evaluate our method in the MM-OSTTA setting.

- The effectiveness and versatility of our approach are validated through extensive experiments across two downstream tasks and five modalities, as well as in challenging *long-term* and *continual* MM-OSTTA scenarios.

## 2 MULTIMODAL OPEN-SET TEST-TIME ADAPTATION

Multimodal Open-set Test-Time Adaptation (MM-OSTTA) aims to adapt a pre-trained source model to a target domain that experiences both ***distribution shifts and label shifts*** across ***multiple modalities***. Let $\mathcal{D}_S = \{(\mathbf{x}_i, y_i)\}_{i=1}^{N_S}$ represent the source domain dataset with label space $\mathcal{C}_S$, which follows the distribution $P_{\mathcal{X}\mathcal{Y}}^S$, where each sample $\mathbf{x}_i$ consists of $M$ modalities, denoted as $\mathbf{x}_i = \{x_i^k \mid k = 1, \cdots, M\}$. Similarly, let $\mathcal{D}_T = \{(\mathbf{x}_i, y_i)\}_{i=1}^{N_T}$ represent the target domain dataset with label space $\mathcal{C}_T$ and distribution $P_{\mathcal{X}\mathcal{Y}}^T$. Let $f : \mathcal{X} \mapsto \mathbb{R}^C$ denote a neural network trained on the source distribution $P_{\mathcal{X}\mathcal{Y}}^S$, where $C$ is the number of classes in $\mathcal{C}_S$. In MM-OSTTA, $f$ consists of $M$ feature extractors $g_k(\cdot)$ and a classifier $h(\cdot)$. Each feature extractor $g_k(\cdot)$ processes modality $k$ to produce an embedding $\mathbf{Z}^k$, and the classifier $h(\cdot)$ combines these embeddings to generate a prediction probability $\hat{p}$:

$$\hat{p} = \delta(f(\mathbf{x})) = \delta(h([g_1(x^1), ..., g_M(x^M)])) = \delta(h([\mathbf{Z}^1, ..., \mathbf{Z}^M])), \tag{1}$$

where $\delta(\cdot)$ denotes the softmax function. Additionally, we include separate classifiers $h_k(\cdot)$ for each modality $k$, yielding modality-specific prediction probabilities $\hat{p}^k = \delta(h_k(g_k(x^k)))$.

Given a well-trained multimodal source model $f(\mathbf{x})$ on $\mathcal{D}_S$, MM-OSTTA aims to adapt this model to the target domain $\mathcal{D}_T$, where $P_{\mathcal{X}\mathcal{Y}}^S \neq P_{\mathcal{X}\mathcal{Y}}^T$. Unlike traditional closed-set TTA, which assumes $\mathcal{C}_S = \mathcal{C}_T$, MM-OSTTA operates under the condition $\mathcal{C}_S \subseteq \mathcal{C}_T$, meaning the target domain may contain samples from *unknown* classes not present in the source domain. In addition to adapting the model and making predictions, MM-OSTTA involves generating a prediction score $S(\mathbf{x})$ for each sample and employing an unknown class detector $G_\eta(x)$, defined as:

$$G_\eta(x) = \begin{cases} \text{known} & S(\mathbf{x}) \geq \eta \\ \text{unknown} & S(\mathbf{x}) < \eta \end{cases}, \tag{2}$$

where $\eta$ is a predefined threshold. Samples with $S(\mathbf{x}) < \eta$ are classified as unknown. We use the Maximum Softmax Probability (MSP) (Hendrycks & Gimpel, 2017) as $S(\mathbf{x})$ by default.

## 3 METHODOLOGY

### 3.1 CORRELATION BETWEEN ENTROPY AND MM-OSTTA PERFORMANCE

We begin by exploring the relationship between prediction entropy, defined as $H(\hat{p}) = -\sum_c \hat{p}_c \log \hat{p}_c$, and the performance of MM-OSTTA. Using a pre-trained model on the source domain, we generate predictions on the target domain without performing any adaptation. The target domain consists of both known and unknown samples. To quantify this relationship, we calculate the average prediction entropy for known ($H_{known}$) and unknown samples ($H_{unknown}$) separately, and then compute the difference ($H_{unknown} - H_{known}$). We evaluate this entropy difference across various domain-shift scenarios using the EPIC-Kitchens (Damen et al., 2018) dataset and analyze its correlation with an MM-OSTTA performance metric (FPR95), which measures the unknown class detection ability. A lower FPR95 indicates better performance. As shown in Fig. 2, there is a strong correlation between the entropy difference and MM-OSTTA performance, with a larger entropy difference corresponding to a lower FPR95. This observation is intuitive, as a higher entropy difference suggests that unknown samples exhibit significantly greater entropy than known samples, making them easier to differentiate.

Tent (Wang et al., 2021) minimizes the entropy for all samples, regardless of whether they belong to known or unknown classes, which inadvertently decreases the entropy difference between these

classes. This reduction in entropy difference leads to diminished performance, as demonstrated in Fig. 2 and Fig. 6. To overcome this limitation and amplify the entropy difference between known and unknown samples during online adaptation, we propose the **Adaptive Entropy-aware Optimization (*AEO*)** framework. *AEO* consists of two primary components: Unknown-aware Adaptive Entropy Optimization (UAE) (Sec. 3.2) and Adaptive Modality Prediction Discrepancy Optimization (AMP) (Sec. 3.3).

### 3.2 UNKNOWN-AWARE ADAPTIVE ENTROPY OPTIMIZATION

As discussed in Sec. 3.1, improving MM-OSTTA performance requires increasing the entropy difference between known and unknown class samples, *i.e.*, maximizing the entropy of unknown samples while minimizing that of known samples. Tent (Wang et al., 2021) minimizes the entropy of all samples, failing to enhance this difference. Some approaches (Niu et al., 2023; Yang et al., 2024) apply entropy minimization selectively to high-confident samples, but still struggle to increase the entropy difference. The first step in effectively optimizing entropy for both known and unknown samples is to reliably identify potential unknown samples. To address this, we introduce the Unknown-aware Adaptive Entropy Optimization (UAE) loss, which adaptively weights and optimizes each sample based on its prediction uncertainty. The UAE loss is defined as:

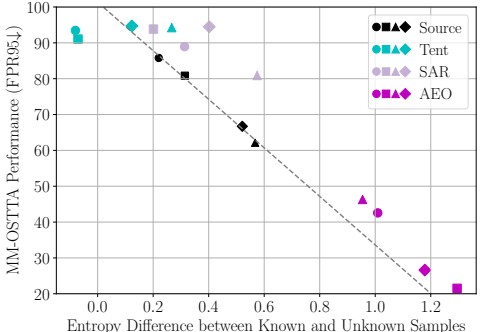

Figure 2: The entropy difference between known and unknown samples is positively correlated with the MM-OSTTA performance. Tent minimizes the entropy of all samples, regardless of whether they are known or unknown, thereby failing to increase entropy differences and leading to poorer performances. In contrast, our *AEO* amplifies entropy differences via adaptive optimization, significantly improving unknown detection. *Different shapes represent different domain-shift scenarios.*

$$W_{ada} = Tanh(\beta \cdot (H(\hat{p}) - \alpha)), \quad (3)$$

$$\mathcal{L}_{AdaEnt} = -H(\hat{p}) \cdot W_{ada}, \quad (4)$$

where $Tanh$ is the hyperbolic tangent function, $W_{ada}$ is the adaptive weight assigned to each sample, $H(\hat{p})$ is the normalized entropy of prediction $\hat{p}$, computed as $H(\hat{p}) = -(\sum_c \hat{p}_c \log \hat{p}_c)/log(C)$, with $C$ being the number of classes. The parameters $\alpha$ and $\beta$ are hyperparameters that control the entropy threshold and scaling, respectively.

The function $Tanh(x)$ is positive when $x > 0$ and negative when $x < 0$. Therefore, the UAE loss $\mathcal{L}_{AdaEnt}$ maximizes $H(\hat{p})$ when $H(\hat{p}) > \alpha$ (*i.e.* when prediction confidence is low, indicating the sample is likely unknown) and minimizes $H(\hat{p})$ when $H(\hat{p}) < \alpha$ (*i.e.* when prediction confidence is high, indicating the sample is likely known). Moreover, $Tanh(x)$ asymptotically approaches 1 as $x$ increases and converges to $-1$ as $x$ decreases, resulting in higher weights for samples with very high or very low $H(\hat{p})$ (*i.e.* those that most likely to be known or unknown). When $H(\hat{p})$ is close to $\alpha$, the model is uncertain about whether the sample is known or unknown, and there is a higher risk of wrong predictions. In such cases, the assigned weight approaches 0, effectively neutralizing the potential negative impact of uncertain samples. In this manner, our UAE loss adaptively optimizes the entropy for each sample, enhancing the separation between known and unknown samples to ensure more reliable predictions. More discussion on the importance of $W_{ada}$ are in Appendix C.13.

### 3.3 ADAPTIVE MODALITY PREDICTION DISCREPANCY OPTIMIZATION

To further enhance the entropy difference between known and unknown samples, we introduce Adaptive Modality Prediction Discrepancy Optimization (AMP), which optimizes the predictions across different modalities. To achieve this, AMP first employs an adaptive entropy loss, similar to the UAE, that increase the entropy of predictions from each modality when confidence is low, and decrease it when confidence is high. The loss is defined as:

$$\mathcal{L}_{AdaEnt*} = -\frac{1}{2}(H(\hat{p}^1) + H(\hat{p}^2)) \cdot W_{ada}, \quad (5)$$

where $W_{ada}$ is the adaptive weight calculated in Eq. (3). Additionally, we propose maximizing the prediction discrepancy between modalities for unknown samples to encourage uncertainy

(i.e., diversifying predictions across modalities increases the uncertainty in the final prediction). Conversely, for known samples, we enforce consistency across modalities to ensure confident predictions (i.e., confident predictions should exhibit consistent outputs across all modalities). To achieve this, we define the adaptive modality prediction discrepancy loss as:

$$\mathcal{L}_{AdaDis} = -(Dis(\hat{p}^1, \hat{p}^2)) \cdot W_{ada}, \qquad (6)$$

where $W_{ada}$ is the adaptive weight from Eq. (3) and $\mathcal{L}_{AdaDis}$ emphasizes samples with either very high or low $H(\hat{p})$. $Dis(\cdot)$ measures the prediction discrepancy between two modalities, with $L_1$ distance being the default choice. As illustrated in Fig. 3, training with AMP further increases the entropy difference, leading to improved performance. We also include a negative entropy loss term $\mathcal{L}_{Div}$ to ensure diversity in predictions (Zhou et al., 2023; Yang et al., 2024):

$$\mathcal{L}_{Div} = \sum_{c=1}^{C} p_c \log p_c, \qquad (7)$$

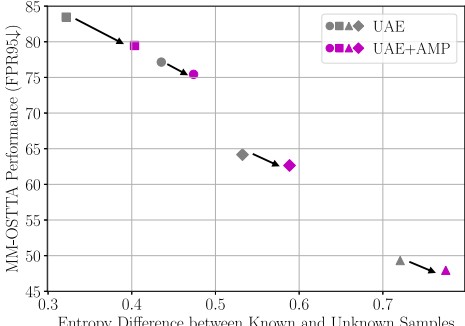

Figure 3: Training with AMP further amplifies the entropy difference between known and unknown samples, leading to improved performances.

where $p_c$ is the accumulated prediction probability for class $c$ over one batch. The final loss is computed as the weighted sum of the previously defined losses:

$$\mathcal{L}_{AEO} = \mathcal{L}_{AdaEnt} + \gamma_1(\mathcal{L}_{AdaEnt*} + \mathcal{L}_{AdaDis}) + \gamma_2\mathcal{L}_{Div}. \qquad (8)$$

As shown in Fig. 2, our *AEO* significantly amplifies the entropy difference between known and unknown samples at test time, resulting in substantial performance improvements.

## 4 EXPERIMENTS

We evaluate our proposed method across four benchmark datasets: EPIC-Kitchens and Human-Animal-Cartoon (HAC) for multimodal **action recognition** with domain shifts, Kinetics-100-C for multimodal action recognition under corruptions, and the nuScenes dataset for multimodal **3D semantic segmentation** in Day-to-Night and USA-Singapore adaptation scenarios.

### 4.1 EXPERIMENT SETTINGS

**Datasets.** For domain adaptation experiments, we utilize the widely adopted EPIC-Kitchens (Damen et al., 2018) and HAC (Dong et al., 2023) datasets. Both datasets offer three modalities: video, audio, and optical flow. The EPIC-Kitchens dataset comprises eight actions ('put', 'take', 'open', 'close', 'wash', 'cut', 'mix', and 'pour') recorded in three distinct kitchens, forming three domains D1, D2, and D3. The HAC dataset includes seven actions ('sleeping', 'watching TV', 'eating', 'drinking', 'swimming', 'running', and 'opening door') performed by humans (H), animals (A), and cartoon (C) figures, resulting in three distinct domains: H, A, and C. In our experiments, models are pre-trained on a source domain and adapted to a target domain online. For the open-set setting, we treat HAC samples as unknown classes for EPIC-Kitchens and vice versa. To prevent class overlap, we exclude the 'open' class samples from EPIC-Kitchens dataset and the 'opening door' class from the HAC dataset when used as unknowns.

For the corruption robustness experiments, models are trained on clean datasets and adapted to corrupted test sets. We create the Kinetics-100-C dataset, which includes video and audio modalities, following the approaches outlined in Hendrycks & Dietterich (2019) and Yang et al. (2024). Kinetics-100-C consists of 100 classes selected from Kinetics-600 dataset (Carreira et al., 2018), with 21181 videos for training and validation, and 3800 videos for testing. We apply six types of corruptions on videos (Gaussian, Defocus, Frost, Brightness, Pixelate, and JPEG) and audios (Gaussian, Wind, Traffic, Thunder, Rain, and Crowd), generating six distinct corruption shifts. For example, *Defocus (v) + Wind (a)* indicates defocus corruption on video and wind corruption on audio. All experiments are conducted under the most severe corruption level 5 (Hendrycks & Dietterich, 2019). For the open-set setting, we utilize HAC as the unknown classes, applying the same corruption types as in

Kinetics-100-C, resulting in the HAC-C dataset. By applying identical corruption types, we create open-set samples from a matching domain shift but with unknown classes.

For multimodal 3D semantic segmentation, we utilize the nuScenes dataset (Caesar et al., 2020), which includes LiDAR and camera modalities. We examine two realistic adaptation scenarios following Jaritz et al. (2020): (1) **Day-to-Night** adaptation: LiDAR exhibits minimal domain shift due to its active sensing capabilities (emitting laser beams that remain largely unaffected by lighting conditions). In contrast, the camera, functioning as a passive sensor, experiences a significant domain gap due to poor light at night, leading to substantial changes in object appearance. (2) **USA-Singapore** country-to-country adaptation: The domain gap may vary for both LiDAR and camera modalities. For certain classes, the 3D shape may shift more significantly than the visual appearance, while for others, the reverse may hold true. In the open-set setting, we designate all vehicle classes as unknown. During training, unknown classes are labeled as void and ignored. During inference, the objective is to segment the known classes while simultaneously detecting unknown classes. Further illustrations and dataset details are provided in Appendix B.4.

**Evaluation Metrics.** To evaluate the model's adaptation performance on known data, we use accuracy (Acc) for classification tasks and mean Intersection over Union (IoU) for segmentation tasks. To assess the model's ability to robustly detect unknown classes, we measure the area under the receiver operating characteristic curve (AUROC) and the false positive rate of unknown samples when the true positive rate at 95% (FPR95) for unknown samples. As our objective is to achieve a good balance between the classification accuracy of known classes and the detection accuracy of unknown classes, we reformulate a novel version of *H-score*, defined as the harmonic mean of Acc, AUROC, and FPR95:

$$H\text{-}score = \frac{3}{\frac{1}{Acc} + \frac{1}{AUROC} + \frac{1}{1-FPR95}}. \tag{9}$$

Since a lower FPR95 indicates better performance, we use $1 - FPR95$ for the H-score calculation in Eq. (9). AUROC provides a global measure of how well the model distinguishes between known and unknown classes across all possible thresholds, making it suitable for tasks requiring balanced performance across thresholds. FPR95 evaluates the model's performance at a specific recall level (95% TPR), which is particularly important in applications requiring high recall, such as fraud detection or outlier detection. To comprehensively evaluate the model under the open-set setting, both FPR95 and AUROC are included in our H-score calculation. For the segmentation task, we replace Acc with IoU to calculate H-score.

**Baseline models.** We compare our method against two unimodal TTA methods, Tent (Wang et al., 2021) and SAR (Niu et al., 2023), as well as two unimodal open-set TTA methods, OSTTA (Lee et al., 2023) and UniEnt (Gao et al., 2024), along with one multimodal TTA method, READ (Yang et al., 2024). Due to space limitations, additional implementation details are provided in Appendix B.2 and Appendix B.3.

## 4.2 COMPARISONS WITH STATE-OF-THE-ART

**Robustness under domain shifts.** We first conduct comprehensive experiments on domain adaptation benchmarks, where a model is trained on a single source domain and then adapted online to a target domain with significant distribution shifts. Tab. 1 presents results from the EPIC-Kitchens dataset using video and audio modalities. Unimodal TTA methods, such as Tent (Wang et al., 2021) and SAR (Niu et al., 2023), underperform in the multimodal open-set TTA setup, revealing their limited adaptability in complex scenarios involving multiple modalities and unknown classes. Similarly, OSTTA (Lee et al., 2023), a unimodal open-set TTA method, struggles to achieve robust performance, highlighting the inherent challenges of multimodal open-set TTA. In contrast, UniEnt (Gao et al., 2024), another unimodal open-set TTA method, performs well in this setup. The SOTA multimodal TTA method READ (Yang et al., 2024) demonstrates competitive performance, improving the Source baseline H-score by 3.55%. READ achieves this by focusing entropy minimization on high-confidence predictions while mitigating the noise from low-confidence ones. Our proposed *AEO* framework demonstrates strong robustness in the challenging open-set setup, significantly improving the Source baseline H-score by 22.07%. Notably, in the D1 → D3 adaptation, *AEO* improves FPR95 metric by a relative value of 59.31%, which is crucial in applications requiring high sensitivity.

| | D1 → D2 | | | | D1 → D3 | | | | D2 → D1 | | | | D2 → D3 | | | |
|---|---|---|---|---|---|---|---|---|---|---|---|---|---|---|---|---|
| | Acc↑ | FPR95↓ | AUROC↑ | H-score↑ | Acc↑ | FPR95↓ | AUROC↑ | H-score↑ | Acc↑ | FPR95↓ | AUROC↑ | H-score↑ | Acc↑ | FPR95↓ | AUROC↑ | H-score↑ |
| Source | 48.54 | 85.79 | 58.31 | 27.75 | 48.31 | 80.83 | 63.09 | 33.82 | 46.92 | 62.13 | 80.06 | 49.83 | 51.57 | 66.72 | 77.20 | 48.08 |
| Tent | 44.04 | 93.47 | 43.45 | 15.09 | 48.96 | 91.05 | 44.55 | 19.40 | 46.06 | 94.24 | 62.63 | 14.20 | 46.03 | 94.72 | 54.86 | 13.08 |
| SAR | 49.12 | 88.94 | 63.50 | 23.71 | 50.30 | 93.84 | 57.58 | 15.03 | 46.63 | 80.89 | 74.38 | 34.40 | 46.11 | 94.48 | 64.96 | 13.75 |
| OSTTA | 47.09 | 97.04 | 33.94 | 7.72 | 50.15 | 95.57 | 39.10 | 11.06 | 48.79 | 90.29 | 64.45 | 21.58 | 44.39 | 89.82 | 67.34 | 22.12 |
| UniEnt | 47.46 | 73.10 | 76.67 | 42.08 | 45.58 | 62.90 | 85.36 | 49.50 | 47.37 | 45.65 | 87.98 | **58.97** | 51.94 | 41.82 | 90.71 | 63.20 |
| READ | 47.73 | 79.63 | 68.48 | 35.44 | 48.72 | 78.46 | 68.69 | 36.81 | 49.90 | 73.36 | 75.94 | 42.41 | 54.20 | 67.67 | 77.87 | 48.21 |
| AEO (Ours) | 50.79 | 42.56 | 90.92 | **62.37** | 48.68 | 21.52 | 96.60 | **68.75** | 46.87 | 46.31 | 89.88 | 58.72 | 53.77 | 26.61 | 94.85 | **70.15** |

| | D3 → D1 | | | | D3 → D2 | | | | Mean | | | |
|---|---|---|---|---|---|---|---|---|---|---|---|---|
| | Acc↑ | FPR95↓ | AUROC↑ | H-score↑ | Acc↑ | FPR95↓ | AUROC↑ | H-score↑ | Acc↑ | FPR95↓ | AUROC↑ | H-score↑ |
| Source | 46.31 | 90.19 | 59.67 | 21.38 | 58.43 | 89.06 | 57.28 | 23.81 | 50.01 | 79.12 | 65.94 | 34.11 |
| Tent | 46.01 | 91.05 | 57.45 | 19.88 | 51.53 | 92.39 | 48.17 | 17.49 | 47.11 | 92.82 | 51.85 | 16.52 |
| SAR | 45.75 | 85.79 | 62.17 | 27.70 | 50.97 | 90.88 | 54.16 | 20.31 | 48.15 | 89.14 | 62.79 | 22.48 |
| OSTTA | 45.35 | 85.54 | 60.41 | 27.84 | 50.88 | 88.75 | 54.31 | 23.63 | 47.78 | 91.17 | 53.26 | 18.99 |
| UniEnt | 49.14 | 85.39 | 65.90 | 28.85 | 58.12 | 87.55 | 63.21 | 26.47 | 49.94 | 66.07 | 78.30 | 44.85 |
| READ | 49.65 | 83.72 | 66.19 | 31.03 | 57.50 | 83.36 | 61.97 | 32.04 | 51.28 | 77.70 | 69.86 | 37.66 |
| AEO (Ours) | 49.14 | 75.43 | 72.79 | **40.11** | 56.33 | 79.48 | 68.43 | **36.99** | 50.93 | 48.65 | 85.58 | **56.18** |

Table 1: Multimodal Open-set TTA with video and audio modalities on EPIC-Kitchens dataset.

| | *Mean* (video+audio) | | | | *Mean* (video+flow) | | | | *Mean* (flow+audio) | | | | *Mean* (video+audio+flow) | | | |
|---|---|---|---|---|---|---|---|---|---|---|---|---|---|---|---|---|
| | Acc↑ | FPR95↓ | AUROC↑ | H-score↑ | Acc↑ | FPR95↓ | AUROC↑ | H-score↑ | Acc↑ | FPR95↓ | AUROC↑ | H-score↑ | Acc↑ | FPR95↓ | AUROC↑ | H-score↑ |
| Source | 56.36 | 78.24 | 56.68 | 34.61 | 57.81 | 75.69 | 58.15 | 37.11 | 44.32 | 79.45 | 63.73 | 33.50 | 56.66 | 76.31 | 61.81 | 37.68 |
| Tent | 59.05 | 85.19 | 59.75 | 28.87 | 58.61 | 86.71 | 57.78 | 26.55 | 44.32 | 89.01 | 55.04 | 21.12 | 58.86 | 86.91 | 53.60 | 25.42 |
| SAR | 60.66 | 78.95 | 63.88 | 35.94 | 60.17 | 85.92 | 59.70 | 27.19 | 43.75 | 82.32 | 61.29 | 27.43 | 60.56 | 82.34 | 59.74 | 31.35 |
| OSTTA | 58.84 | 82.46 | 63.05 | 32.58 | 58.45 | 86.45 | 60.95 | 25.85 | 41.73 | 88.26 | 55.17 | 21.63 | 58.52 | 84.38 | 57.29 | 29.13 |
| UniEnt | 59.34 | 75.59 | 63.35 | 37.48 | 60.09 | 71.51 | 66.02 | 41.34 | 44.62 | 75.17 | 68.90 | 37.60 | 58.23 | 72.13 | 68.25 | 42.03 |
| READ | 58.78 | 72.83 | 66.04 | 42.71 | 59.03 | 74.96 | 66.05 | 40.17 | 43.51 | 76.82 | 66.14 | 36.28 | 57.47 | 74.26 | 67.28 | 41.32 |
| AEO (Ours) | 59.53 | 66.75 | 72.50 | **48.31** | 59.38 | 65.75 | 74.96 | **49.23** | 44.29 | 69.44 | 72.84 | **42.62** | 59.76 | 66.88 | 72.82 | **48.50** |

Table 2: Multimodal Open-set TTA with different combinations of video, audio, and optical flow modalities on HAC dataset.

To assess the generalizability of our proposed method, we further evaluate it on the HAC dataset using different modality combinations: *video+audio*, *video+flow*, *flow+audio*, and *video+audio+flow*, as presented in Tab. 2. For each modality combination, we consider six adaptation scenarios: H → A, H → C, A → H, A → C, C → H, C → A. The results are averaged over six splits (detailed results are available from Tab. 20 to Tab. 23). Consistent with our findings on the EPIC-Kitchens dataset, most existing TTA methods struggle to generalize effectively in the challenging multimodal open-set TTA setup. While UniEnt (Gao et al., 2024) and READ (Yang et al., 2024) perform well and surpass the Source baseline, other TTA methods fail to achieve robust performance, underscoring the complexities of multimodal open-set TTA. In contrast, our method demonstrates strong robustness across all modality combinations, significantly improving the Source baseline H-score by 13.70%, 12.12%, 9.12%, and 10.82% for the respective modality setups. This showcases the effectiveness of our approach in handling diverse multimodal scenarios under challenging open-set conditions.

**Robustness under corruption.** We evaluate our method on the challenging Kinetics-100-C corruption benchmark, which introduces various corruptions to both video and audio modalities. The results are summarized in Tab. 3. The pre-trained model struggles to generalize to corruptions such as *Gaussian (v) + Gaussian (a)* and *Frost (v) + Traffic (a)*, leading to very low H-scores. In contrast, our proposed *AEO* adapts effectively to these corruptions in an online manner, improving the H-score over the Source baseline by 32.78% and 26.50% respectively. Conversely, methods like Tent (Wang et al., 2021) and SAR (Niu et al., 2023) exhibit severe performance degradation under these corruptions. Our *AEO* consistently demonstrates robustness across all types of corruptions, achieving an average H-score improvement of 22.41% over the Source baseline.

**Long-term MM-OSTTA.** Models deployed in real-world scenarios continuously encounter test samples over extended periods and must make reliable predictions at any time. Recent work by Lee et al. (2023) shows that most existing TTA methods perform poorly in long-term settings, often degrading to performance levels worse than non-updating models. Following the methodology of Lee et al. (2023), we simulate long-term TTA by repeating the adaptation process for 10 rounds without resetting the model. The results, summarized in Tab. 4, show a 7.67% increase in H-score after long-term adaptation using our *AEO*. This improvement demonstrates that our method is robust against error accumulation and its ability to continuously optimize the entropy difference between known and unknown samples (Fig. 4). In contrast, most of the baseline methods suffer from significant performance degradation during long-term adaptation.

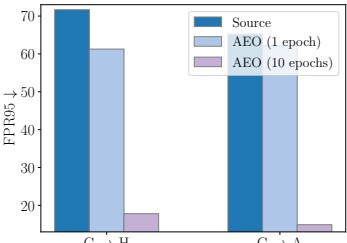

Figure 4: *AEO* continuously optimizes entropy difference between known and unknown samples, resulting in a substantial reduction of FPR95 after 10 adaptation epochs.

| | Defocus (v) + Wind (a) | | | | Frost (v) + Traffic (a) | | | | Brightness (v) + Thunder (a) | | | | Pixelate (v) + Rain (a) | | | |
|---|---|---|---|---|---|---|---|---|---|---|---|---|---|---|---|---|
| | Acc↑ | FPR95↓ | AUROC↑ | H-score↑ | Acc↑ | FPR95↓ | AUROC↑ | H-score↑ | Acc↑ | FPR95↓ | AUROC↑ | H-score↑ | Acc↑ | FPR95↓ | AUROC↑ | H-score↑ |
| Source | 60.74 | 72.63 | 71.93 | 44.84 | 33.24 | 87.68 | 55.33 | 23.20 | 78.97 | 59.50 | 79.13 | 60.01 | 70.63 | 72.11 | 69.38 | 46.56 |
| Tent | 62.24 | 85.87 | 61.09 | 29.07 | 50.21 | 96.26 | 49.48 | 9.76 | 78.05 | 89.63 | 59.13 | 23.78 | 74.66 | 92.18 | 53.82 | 18.77 |
| SAR | 63.32 | 89.45 | 64.89 | 23.81 | 52.13 | 95.71 | 54.57 | 11.09 | 76.76 | 91.42 | 61.84 | 20.58 | 74.82 | 93.37 | 55.33 | 16.46 |
| OSTTA | 62.76 | 81.39 | 65.19 | 35.29 | 51.42 | 86.45 | 61.65 | 27.41 | 76.11 | 87.92 | 67.70 | 27.10 | 75.05 | 79.29 | 72.57 | 39.79 |
| UniEnt | 63.53 | 62.50 | 80.20 | 54.67 | 50.26 | 79.39 | 71.30 | 36.39 | 77.13 | 46.08 | 87.65 | 69.90 | 74.79 | 56.58 | 84.14 | 62.13 |
| READ | 64.24 | 59.08 | 80.19 | 57.17 | 54.95 | 67.47 | 75.62 | 48.26 | 77.74 | 42.21 | 87.82 | 72.19 | 75.63 | 45.32 | 87.03 | 69.77 |
| AEO (Ours) | 63.47 | 54.37 | 83.12 | **60.36** | 54.82 | 65.82 | 77.74 | **49.70** | 77.42 | 40.42 | 89.30 | **73.35** | 75.68 | 42.79 | 88.67 | **71.48** |
| | JPEG (v) + Crowd (a) | | | | Gaussian (v) + Gaussian (a) | | | | *Mean* | | | | | | | |
| | Acc↑ | FPR95↓ | AUROC↑ | H-score↑ | Acc↑ | FPR95↓ | AUROC↑ | H-score↑ | Acc↑ | FPR95↓ | AUROC↑ | H-score↑ | | | | |
| Source | 61.11 | 70.11 | 69.38 | 46.70 | 13.05 | 98.18 | 43.90 | 4.62 | 52.96 | 76.70 | 64.84 | 37.66 | | | | |
| Tent | 70.13 | 93.11 | 53.47 | 16.84 | 35.82 | 97.26 | 49.38 | 7.26 | 61.85 | 92.38 | 54.40 | 17.58 | | | | |
| SAR | 70.05 | 93.71 | 58.22 | 15.75 | 39.08 | 97.16 | 54.57 | 7.58 | 62.69 | 93.47 | 58.24 | 15.88 | | | | |
| OSTTA | 69.79 | 85.24 | 67.46 | 30.96 | 40.47 | 93.82 | 59.63 | 14.76 | 62.60 | 85.69 | 65.70 | 29.22 | | | | |
| UniEnt | 69.53 | 68.18 | 79.67 | 51.40 | 39.18 | 98.61 | 56.02 | 3.93 | 62.40 | 68.56 | 76.50 | 46.40 | | | | |
| READ | 71.58 | 48.42 | 84.76 | 66.44 | 42.97 | 74.95 | 69.41 | **38.66** | 64.52 | 56.24 | 80.80 | 58.75 | | | | |
| AEO (Ours) | 71.05 | 45.87 | 87.12 | **68.14** | 40.82 | 75.53 | 67.34 | 37.40 | 63.88 | 54.13 | 82.22 | **60.07** | | | | |

Table 3: Multimodal Open-set TTA with video and audio modalities on Kinetics-100-C, with corrupted video and audio modalities (severity level 5).

| | *Mean* | | | |
|---|---|---|---|---|
| | Acc↑ | FPR95↓ | AUROC↑ | H-score↑ |
| Source | 56.36 | 78.24 | 56.68 | 34.61 |
| Tent | 56.11 | 97.02 | 47.82 | 7.68 |
| SAR | 57.68 | 97.40 | 45.60 | 6.87 |
| OSTTA | 58.36 | 88.00 | 63.22 | 23.37 |
| UniEnt | 50.91 | 80.27 | 59.85 | 30.07 |
| READ | 53.83 | 77.10 | 61.17 | 36.97 |
| AEO (Ours) | 56.81 | 50.53 | 79.64 | **55.98** |

Table 4: **Long-term** Multimodal Open-set TTA on HAC dataset.

| | *Mean* (HAC) | | | | *Mean* (Kinetics-100-C) | | | |
|---|---|---|---|---|---|---|---|---|
| | Acc↑ | FPR95↓ | AUROC↑ | H-score↑ | Acc↑ | FPR95↓ | AUROC↑ | H-score↑ |
| Source | 56.36 | 78.24 | 56.68 | 34.61 | 52.96 | 76.70 | 64.84 | 37.66 |
| Tent | 55.48 | 88.43 | 54.79 | 23.13 | 52.08 | 94.97 | 42.70 | 11.74 |
| SAR | 58.81 | 87.81 | 58.92 | 23.89 | 54.79 | 95.47 | 36.42 | 10.89 |
| OSTTA | 55.55 | 84.36 | 61.06 | 29.77 | 57.53 | 92.36 | 50.62 | 17.05 |
| UniEnt | 58.56 | 80.68 | 62.52 | 32.30 | 59.62 | 68.33 | 77.99 | 46.43 |
| READ | 57.76 | 73.49 | 65.00 | 41.85 | 61.10 | 48.29 | 84.49 | 62.32 |
| AEO (Ours) | 60.08 | 60.38 | 78.89 | **53.84** | 58.74 | 42.62 | 87.22 | **64.19** |

Table 5: **Continual** Multimodal Open-set TTA on HAC and Kinetics-100-C (severity level 5) datasets.

| | Day → Night | | | | USA → Singapore | | | |
|---|---|---|---|---|---|---|---|---|
| | IoU↑ | FPR95↓ | AUROC↑ | H-score↑ | IoU↑ | FPR95↓ | AUROC↑ | H-score↑ |
| Source | 41.76 | 47.97 | 79.11 | 53.76 | 54.27 | 49.38 | 83.49 | 59.81 |
| Tent | 41.51 | 50.50 | 79.86 | 52.80 | 47.92 | 46.75 | 82.78 | 58.00 |
| READ | 40.32 | 47.03 | 81.84 | 53.67 | 50.09 | 46.39 | 82.56 | 59.14 |
| MM-TTA | 39.98 | 52.86 | 79.30 | 50.99 | 51.42 | 44.13 | 83.43 | 60.87 |
| AEO (Ours) | 42.04 | 44.90 | 82.56 | **55.51** | 55.04 | 41.11 | 84.57 | **63.87** |

Table 6: Multimodal Open-set TTA with **LiDAR** and **camera** modalities on nuScenes dataset.

**Continual MM-OSTTA.** Real-world machine perception systems operate in environments where the target domain distribution evolves continuously. Recent work by Wang et al. (2022b) has highlighted that most existing TTA methods perform poorly in continual settings. Following the setup outlined in Wang et al. (2022b), we simulate continual TTA by sequentially adapting the model across changing domains (*e.g.*, H → A → C for the HAC dataset and similarly for Kinetics-100-C) without resetting the model. The results, presented in Tab. 5 (detailed results are in Tab. 18 and Tab. 19), demonstrate that our method maintains robust performance under this challenging setup, even improving the H-score. In contrast, most baseline methods, particularly unimodal TTA methods (Wang et al., 2021; Niu et al., 2023), exhibit significant performance degradation.

**Scaling to segmentation task.** To further demonstrate the versatility of our *AEO* method beyond action recognition task involving video, audio, and optical flow modalities, we conduct experiments on a novel 3D semantic segmentation task that utilizes LiDAR and camera modalities. As shown in Tab. 6, baseline methods such as READ (Yang et al., 2024) and MM-TTA (Shin et al., 2022) struggle to outperform the source model. In contrast, our *AEO* method consistently demonstrates strong open-set performance, improving FPR95 with up to 8.22% over the Source baseline. This significant improvement highlights the effectiveness of *AEO* in maintaining robust performance across diverse tasks and modalities, underscoring its potential for reliable deployment in real-world segmentation applications.

## 4.3 Ablation Studies and Analysis

**Ablation on each proposed module.** We conducted comprehensive ablation studies to evaluate the contribution of each proposed module, as detailed in Tab. 7. The results indicate that incorporating UAE effectively increases the entropy difference between known and unknown samples, thereby enhancing detection performance for unknown classes. Additionally, integrating AMP maximizes the prediction discrepancy between different modalities for unknown classes, fostering uncertainty in predictions and further improving unknown class detection. These two modules are complementary, and their combined implementation achieves the highest performance across all datasets.

| UAE | AMP | HAC | | | | EPIC-Kitchens | | | | Kinetics-100-C | | | |
|---|---|---|---|---|---|---|---|---|---|---|---|---|---|
| | | Acc↑ | FPR95↓ | AUROC↑ | H-score↑ | Acc↑ | FPR95↓ | AUROC↑ | H-score↑ | Acc↑ | FPR95↓ | AUROC↑ | H-score↑ |
| | | 56.36 | 78.24 | 56.68 | 34.61 | 50.01 | 79.12 | 65.94 | 34.11 | 52.96 | 76.70 | 64.84 | 37.66 |
| ✓ | | 59.76 | 67.58 | 71.53 | 47.79 | 51.07 | 50.12 | 81.30 | 55.01 | 63.49 | 55.44 | 81.30 | 59.01 |
| | ✓ | 56.74 | 69.33 | 70.86 | 45.49 | 45.79 | 63.08 | 80.74 | 48.03 | 58.81 | 56.59 | 79.90 | 56.38 |
| ✓ | ✓ | 59.53 | 66.75 | 72.50 | **48.31** | 50.93 | 48.65 | 85.58 | **56.18** | 63.88 | 54.13 | 82.22 | **60.07** |

Table 7: Ablation on each proposed module with video and audio modalities.

| | Mean (I3D+TSN) | | | |
|---|---|---|---|---|
| | Acc↑ | FPR95↓ | AUROC↑ | H-score↑ |
| Source | 54.29 | 74.63 | 62.14 | 38.39 |
| Tent | 59.80 | 85.52 | 59.18 | 27.58 |
| SAR | 60.08 | 82.67 | 62.86 | 30.40 |
| OSTTA | 59.04 | 84.42 | 63.06 | 28.41 |
| UniEnt | 57.60 | 69.47 | 70.03 | 42.75 |
| READ | 58.00 | 73.48 | 66.89 | 40.89 |
| AEO (Ours) | 59.54 | 66.87 | 74.14 | **47.62** |

Table 8: Ablation on **different architectures** using video and flow modalities on HAC dataset.

| | Mean (SAM) | | | | Mean (SimMMDG) | | | |
|---|---|---|---|---|---|---|---|---|
| | Acc↑ | FPR95↓ | AUROC↑ | H-score↑ | Acc↑ | FPR95↓ | AUROC↑ | H-score↑ |
| Source | 60.25 | 79.54 | 55.50 | 33.87 | 58.87 | 78.93 | 51.23 | 33.77 |
| Tent | 61.24 | 84.06 | 59.04 | 29.77 | 60.27 | 80.79 | 60.44 | 34.47 |
| SAR | 62.23 | 77.80 | 64.06 | 35.78 | 60.54 | 78.72 | 63.02 | 36.20 |
| OSTTA | 61.19 | 77.49 | 63.72 | 38.24 | 59.80 | 77.87 | 63.59 | 36.59 |
| UniEnt | 61.32 | 76.42 | 63.97 | 36.90 | 62.44 | 70.22 | 65.53 | 44.82 |
| READ | 60.48 | 71.74 | 67.26 | 43.86 | 59.93 | 70.16 | 67.62 | 45.71 |
| AEO (Ours) | 60.63 | 67.06 | 72.13 | **48.20** | 60.93 | 66.26 | 70.86 | **49.16** |

Table 9: Ablation on **different pre-trained models** using video and audio modalities on HAC dataset.

**Ablation on hyperparameters in $W_{ada}$.** We evaluate the sensitivity of our method to the hyperparameters in $W_{ada}$ by varying one hyperparameter at a time while keeping the others fixed. Our findings, illustrated in Fig. 5, demonstrate that our method consistently outperforms the best baseline, READ, across all parameter settings. These results indicate that our approach is robust and less sensitive to variations in hyperparameter choices.

**Applicability on different architectures.** To demonstrate the robustness of our *AEO* method across different network architectures, we conducted experiments by modifying the backbone networks. Specifically, we replaced the video backbone with Inflated 3D ConvNet (I3D) (Carreira & Zisserman, 2017) and the optical flow backbone with the Temporal Segment Network (TSN) (Wang et al., 2016). As illustrated in Tab. 8, *AEO* consistently achieves significant performance improvements in MM-OSTTA across these alternative architectures. This consistency underscores the versatility and effectiveness of our approach, regardless of the underlying network design.

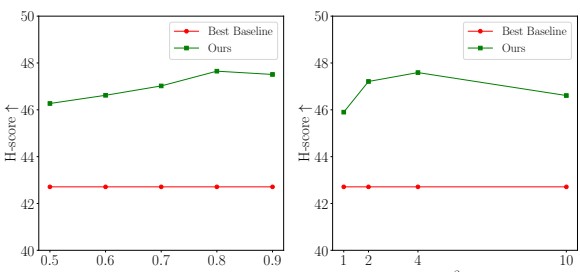

Figure 5: Parameter sensitivity analysis using video and audio modalities on the HAC dataset.

**Robustness to different pre-trained models.** In previous experiments, we pre-trained the model using the standard cross-entropy loss on the training set of each dataset. To assess the robustness of our *AEO* method across various training strategies, we also pre-trained models using advanced optimization techniques such as Sharpness-aware Minimization (SAM) (Foret et al., 2020) and SimMMDG (Dong et al., 2023). As demonstrated in Tab. 9, our method successfully adapts to these differently pre-trained models, consistently achieving the best performance in all configurations.

**Prediction score distributions before and after TTA.** Fig. 6 illustrates the prediction score distributions for known and unknown classes generated by various baseline methods on the EPIC-Kitchens dataset, both before and after applying TTA. Without TTA (Fig. 6 (a)), the score distributions of known and unknown samples significantly overlap, resulting in poor performance in detecting unknown classes. Tent (Wang et al., 2021) minimizes the entropy of all samples indiscriminately, and fails to reduce the separation between known and unknown distributions (Fig. 6 (b)). In contrast, our method achieves better separation between the score distributions of known and unknown classes (Fig. 6 (c)), leading to improved performance in unknown class detection. Fig. 7 illustrates the model prediction entropy for known and unknown samples during online adaptation. Our *AEO* continuously optimizes the entropy batch after batch to improve the MM-OSTTA performance.

**Different ratios of unknown samples.** In real-world scenarios, the proportion of unknown samples can vary significantly. To evaluate how this variability impacts our method's performance, we conducted experiments on the HAC dataset (A → H) using video and audio modalities. Specifically, we adjusted the proportion of unknown class samples in each batch, ranging from 20% to 80%, and present the corresponding H-scores

| | 0.2 | 0.4 | 0.5 | 0.6 | 0.8 |
|---|---|---|---|---|---|
| Source | 20.83 | 41.77 | 57.68 | 42.27 | 19.87 |
| UniEnt | 53.28 | 63.49 | 53.46 | 58.07 | 40.90 |
| READ | 37.23 | 58.37 | 57.76 | 59.49 | 55.50 |
| AEO (Ours) | **53.78** | **68.18** | **65.26** | **67.77** | **69.58** |

Table 10: Ablation on different ratios of unknown samples.

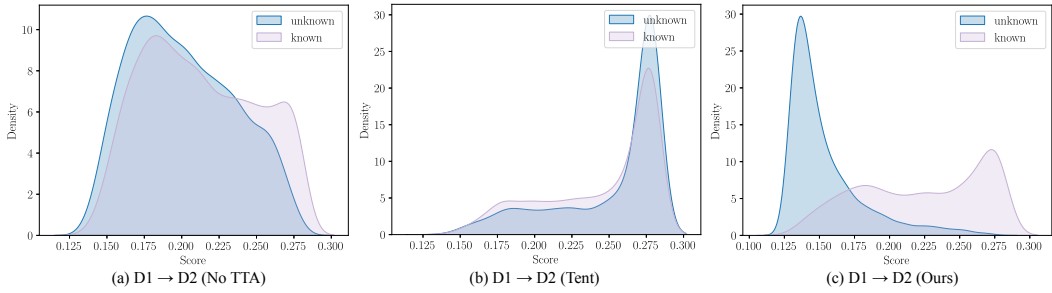

Figure 6: Prediction score distributions of different baseline methods on the EPIC-Kitchens dataset before and after TTA. *AEO* achieves better separation between the score distributions of known and unknown classes, leading to improved performance in unknown class detection.

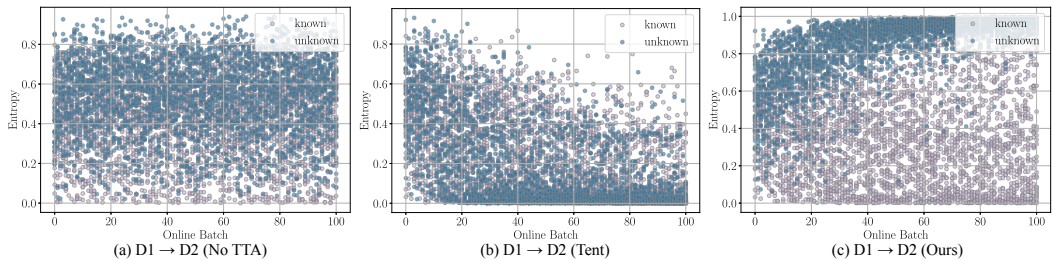

Figure 7: Model prediction entropy for known and unknown samples during online adaptation. Tent fails to reduce the separation between known and unknown distribution. In contrast, our *AEO* continuously optimizes the entropy batch after batch to improve the MM-OSTTA performance.

in Tab. 10. The results indicate that our method remains robust across different proportions of unknown class samples, consistently achieving the highest H-scores in all scenarios.

**Robustness under mixed distribution shifts.** In this setup, the test data originate from multiple shifted domains that are mixed together during adaptaion (Niu et al., 2023), complicating the problem further. As shown in Tab. 11, our *AEO* consistently achieves the best performance in terms of H-score, suggesting its effectiveness under different challenging scenarios. In contrast, several baseline methods, such as Tent (Wang et al., 2021), SAR (Niu et al., 2023), and UniEnt (Gao et al., 2024) suffer from significant performance degradation.

|  | *Mean* | | | |
|---|---|---|---|---|
|  | Acc↑ | FPR95↓ | AUROC↑ | H-score↑ |
| Source | 56.36 | 78.24 | 56.68 | 34.61 |
| Tent | 58.27 | 91.34 | 55.75 | 19.88 |
| SAR | 57.62 | 89.43 | 58.40 | 22.53 |
| OSTTA | 57.56 | 81.33 | 64.40 | 33.17 |
| UniEnt | 57.92 | 81.64 | 59.48 | 29.27 |
| READ | 56.67 | 73.15 | 66.17 | 42.78 |
| AEO (Ours) | 58.87 | 65.54 | 74.52 | **49.85** |

Table 11: Ablation under *mixed* distribution shifts on HAC dataset using video and audio.

## 5  CONCLUSION

In this work, we tackle the challenging task of Multimodal Open-set Test-time Adaptation (MM-OSTTA) for the first time. Motivated by the observation that the entropy difference between *known* and *unknown* class samples positively correlates with MM-OSTTA performance, we propose Adaptive Entropy-aware Optimization (*AEO*). *AEO* consists of two key components: Unknown-aware Adaptive Entropy Optimization (UAE) and Adaptive Modality Prediction Discrepancy Optimization (AMP). Together, these components increase the entropy difference between known and unknown class samples during online adaptation in a complementary manner. We conduct extensive experiments on the newly introduced benchmark, encompassing two downstream tasks and five different modalities, to demonstrate the efficacy and versatility of our proposed *AEO*. Furthermore, *AEO* achieves promising results in both *long-term* and *continual* MM-OSTTA settings, where the adaptation process is repeated over multiple rounds and the target domain distribution evolves over time. These results underscore the robustness and adaptability of *AEO* in dynamic, real-world environments.

## ACKNOWLEDGMENTS

The authors acknowledge the support of "In-service diagnostics of the catenary/pantograph and wheelset axle systems through intelligent algorithms" (SENTINEL) project, supported by the ETH Mobility Initiative.

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

# A    RELATED WORK

## A.1    TEST-TIME ADAPTATION

Test-time Adaptation (TTA) seeks to adapt a pre-trained model on the source domain online, addressing distribution shifts without requiring access to either source data or target labels. This characteristic distinguishes TTA from domain generalization (Wang et al., 2022a) and domain adaptation (Wang & Deng, 2018), giving it broader applicability (Liang et al., 2024). Online TTA methods (Wang et al., 2021; Yuan et al., 2023) update specific model parameters using incoming test samples based on unsupervised objectives such as entropy minimization and pseudo-labels. Robust TTA methods (Niu et al., 2022; Zhou et al., 2023) address more complex and practical scenarios, including label shifts, single-sample adaptation, and mixed domain shifts. Continual TTA approaches (Wang et al., 2022b; Gan et al., 2023) target the continual and evolving distribution shifts encountered over test time. Additionally, several TTA techniques have been developed for specific tasks such as semantic segmentation (Shin et al., 2022) and action recognition (Yang et al., 2024), often involving multiple modalities.

## A.2    OPEN-SET TEST-TIME ADAPTATION

Open-set test-time adaptation (OSTTA) addresses situations where the target domain includes classes absent in the source domain, presenting greater challenges due to the risk of incorrect adaptation to unknown class samples, which can cause a significant drop in performance. OSTTA (Lee et al., 2023) mitigates this by filtering out samples with lower confidence in the adapted model than in the original model. UniEnt (Gao et al., 2024) distinguishes between pseudo-known and pseudo-unknown samples in the test data, applying entropy minimization to the pseudo-known data and entropy maximization to the pseudo-unknown data. STAMP (Yu et al., 2024) optimizes over a stable memory bank rather than the risky mini-batch, where the memory bank is dynamically updated by selecting low-entropy and label-consistent samples in a class-balanced manner. However, existing open-set TTA methods focus exclusively on unimodal settings, overlooking the complexities of multimodal scenarios.

## A.3    MULTIMODAL ADAPTATION AND GENERALIZATION

Multimodal domain adaptation (DA) and domain generalization (DG) tackle the challenge of distribution shifts in multiple modalities, such as video and audio. For instance, MM-SADA (Munro & Damen, 2020) proposes a self-supervised alignment method combined with adversarial alignment for multimodal DA. Similarly, Kim et al. (2021) employ cross-modal contrastive learning to align representations across both modalities and domains. Besides, Zhang et al. (2022) introduce an audio-adaptive encoder and an audio-infused recognizer to mitigate domain shifts. RNA-Net (Planamente et al., 2022) addresses the multimodal DG problem by introducing a relative norm alignment loss that balances the feature norms of audio and video modalities. SimMMDG (Dong et al., 2023) presents a universal framework for multimodal domain generalization by separating features within each modality into modality-specific and modality-shared components, while applying constraints to encourage meaningful representation learning. Building on SimMMDG, MOOSA (Dong et al., 2024a) introduces two self-supervised tasks, Masked Cross-modal Translation and Multimodal Jigsaw Puzzles, to simultaneously address multimodal open-set domain generalization and adaptation. Unlike existing multimodal DA and DG methods, which typically aim to train models using both labeled source domain data and unlabeled target domain data simultaneously or to develop more generalizable neural networks from the source domain, our work focuses on improving the performance of pre-trained multimodal models during test-time by leveraging unlabeled online data from the target domain.

## A.4    OUT-OF-DISTRIBUTION DETECTION

Out-of-Distribution (OOD) detection aims to identify test samples that exhibit semantic shifts without compromising in-distribution (ID) classification accuracy. Numerous OOD detection algorithms have been developed, which can be broadly categorized into post hoc methods and training-time regularization (Yang et al., 2022). *Post hoc* methods design OOD scores based on the classification outputs of neural networks, offering the advantage of ease of use without modifying the training procedure or objective. Popular OOD scores include Maximum Softmax Probability (MSP) (Hendrycks &

Gimpel, 2017), MaxLogit (Hendrycks et al., 2022), Energy (Liu et al., 2020), ReAct (Sun et al., 2021), ASH (Djurisic et al., 2022), Mahalanobis (Lee et al., 2018), $k$-Nearest Neighbor (KNN) (Sun et al., 2022), etc. *Training-time regularization* methods, such as LogitNorm (Wei et al., 2022), address prediction overconfidence by imposing a constant vector norm on the logits during training. In contrast, Outlier Exposure (Hendrycks et al., 2019) uses external OOD samples from other datasets during training to improve discrimination between ID and OOD samples. Additionally, Du et al. (2022) propose synthesizing virtual outliers for training-time regularization. While most existing OOD methods are designed for unimodal scenarios, a recent work (Dong et al., 2024b) introduces the first benchmark for multimodal OOD detection (Li et al., 2024) along with the Agree-to-Disagree algorithm, which enhances multimodal OOD detection performance. However, traditional OOD detection methods assume that both training and test data originate from the same domain – an unrealistic assumption in real-world conditions and TTA scenarios. In realistic settings, domain shifts between training and test data pose additional challenges for OOD detection.

# B   FURTHER IMPLEMENTATION DETAILS

## B.1   PSEUDO CODE

This section presents the pseudo-code for our *AEO* method. From Algorithm 1, test samples are coming batch by batch. For each sample, we first compute the prediction probability $\hat{p}$ from all modalities as well as $\hat{p}^k$ from each modality $k$. We then obtain an initial prediction $\hat{y}$ from $\hat{p}$ and calculate an adaptive weight $W_{ada}$ for each sample. Next, we compute the losses from UAE and AMP to formulate the final loss $\mathcal{L}_{AEO}$ and update model parameters using Adam optimizer. Finally, we output a prediction as well as a corresponding score for each sample. Samples with scores below a predefined threshold will be treated as unknown classes.

---

**Algorithm 1** The Pipeline of *AEO*

---

**Input:** Unlabeled test samples $\mathcal{D}_T = \{\mathbf{x}_i\}_{i=1}^{N_T}$, a pre-trained multimodal model $f_{\theta_t}(\cdot)$.
**for** each online mini-batch $X_b$ **do**
    Calculate $\hat{p}$ and $\hat{p}^k$ for all $\mathbf{x} \in X_b$.
    Obtain the prediction $\hat{y} = \arg\max_c[\hat{p}]_c$ for all $\mathbf{x} \in X_b$.
    Obtain the adaptive weight $W_{ada}$ for all $\mathbf{x} \in X_b$ by Eq. (3).
    Compute $\mathcal{L}_{AdaEnt}$ in UAE by Eq. (4).
    Compute $\mathcal{L}_{AdaEnt*}$ and $\mathcal{L}_{AdaDis}$ in AMP by Eq. (5) and Eq. (6).
    Compute final loss $\mathcal{L}_{AEO}$ using Eq. (8).
    Update $\theta_t$ with Adam optimizer.
**end for**
**Output:** The prediction and score $\{\hat{y}_i, \max_c(\hat{p}_c(\mathbf{x}_i))\}_{i=1}^{N_T}$.

---

## B.2   IMPLEMENTATION DETAILS ON ACTION RECOGNITION TASK

For the action recognition task, we conduct experiments across three modalities: video, audio, and optical flow. We use the SlowFast network (Feichtenhofer et al., 2019) to encode video data and ResNet-18 (He et al., 2016) for audio, and the SlowFast network's slow-only pathway for optical flow. The models are pre-trained on each dataset's training set using standard cross-entropy loss. The Adam optimizer (Kingma & Ba, 2015) is employed with a learning rate of $0.0001$ and a batch size of $16$. Training is performed for $20$ epochs on an RTX 3090 GPU, and the model with the best validation performance is selected. We also pre-train models using advanced training strategies such as Sharpness-aware Minimization (SAM) (Foret et al., 2020) and SimMMDG (Dong et al., 2023), to evaluate TTA performance on different models. During open-set TTA, we construct mini-batches with equal numbers of known and unknown samples. We use a batch size of $64$ and the Adam optimizer with a learning rate of $2e\text{-}5$ for all experiments. We update the parameters of the last layer in each modality's feature encoder as well as the final classification layer. To ensure fairness, we update the same number of parameters for all baseline models. For hyperparameters in $W_{ada}$, we set $\alpha$ to $0.8$ and $\beta$ to $4.0$. For hyperparameters in the final loss $\mathcal{L}_{AEO}$, we set both $\gamma_1$ and $\gamma_2$ to $0.1$.

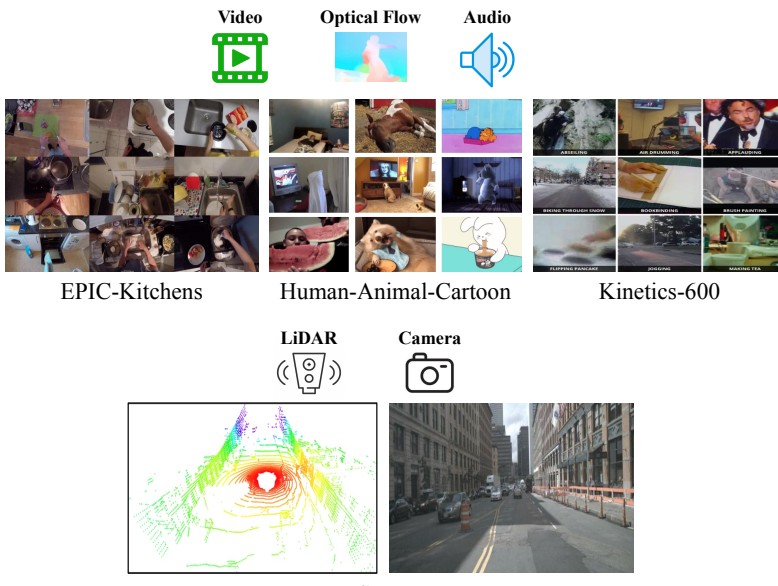

Figure 8: Illustrations of datasets used in our benchmark. We include three action recognition datasets with video, optical flow, and audio modalities, as well as one 3D semantic segmentation dataset with LiDAR and camera modalities.

### B.3 IMPLEMENTATION DETAILS ON SEGMENTATION TASK

For the 3D semantic segmentation task, we use ResNet-34 (He et al., 2016) as the backbone for the camera stream and SalsaNext (Cortinhal et al., 2020) for the LiDAR stream. We adopt the fusion framework proposed in PMF (Zhuang et al., 2021), modifying it by adding an additional segmentation head to the combined features from the camera and LiDAR streams. For optimization, we use SGD with Nesterov (Loshchilov & Hutter, 2016) for the camera stream and Adam (Kingma & Ba, 2015) for the LiDAR stream. The networks are trained for $50$ epochs with a batch size of $3$, starting with a learning rate of $0.0005$, which decays to $0$ with a cosine schedule. In the open-set setting, all vehicle classes are treated as unknown. During training, unknown classes are labeled as void and ignored. To prevent overfitting, we apply various data augmentation techniques, including random horizontal flipping, random scaling, color jitter, 2D random rotation, and random cropping. During open-set TTA, we use a batch size of $1$ and AdamW optimizer with a learning rate of $0.0001$. We update the batch normalization parameters of both LiDAR and camera streams.

### B.4 MORE DETAILS ON BENCHMARK DATASETS

To comprehensively evaluate our proposed methods in the MM-OSTTA setting, we establish a new benchmark derived from existing datasets. This benchmark includes two downstream tasks – action recognition and 3D semantic segmentation – and incorporates five different modalities: video, audio, and optical flow for action recognition, as well as LiDAR and camera for 3D semantic segmentation, as shown in Fig. 8.

**EPIC-Kitchens (Damen et al., 2018).** EPIC-Kitchens is a large-scale egocentric dataset collected from 32 participants in their native kitchen environments. The participants recorded all their daily kitchen activities, with annotated start and end times for each action. We use a subset of the EPIC-Kitchens dataset introduced in the Multimodal Domain Adaptation work (Munro & Damen, 2020), which includes $10,094$ video clips across eight actions ('put', 'take', 'open', 'close', 'wash', 'cut', 'mix', and 'pour'), recorded in three distinct kitchens, forming three separate domains D1, D2, and D3. We use the provided *video* and *audio* modalities.

**HAC (Dong et al., 2023).** The HAC dataset, designed for multimodal domain generalization, features seven actions ('sleeping', 'watching tv', 'eating', 'drinking', 'swimming', 'running', and 'opening door') performed by humans, animals, and cartoon figures, spanning three domains H, A, and C. It

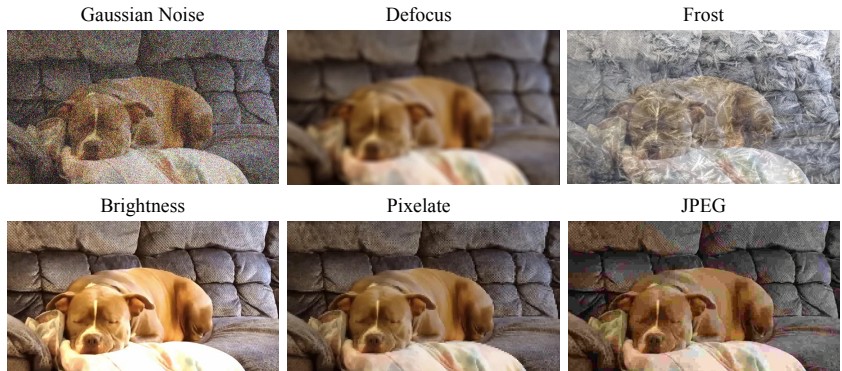

Figure 9: Visualization of various visual corruption types on the constructed Kinetics-100-C benchmark.

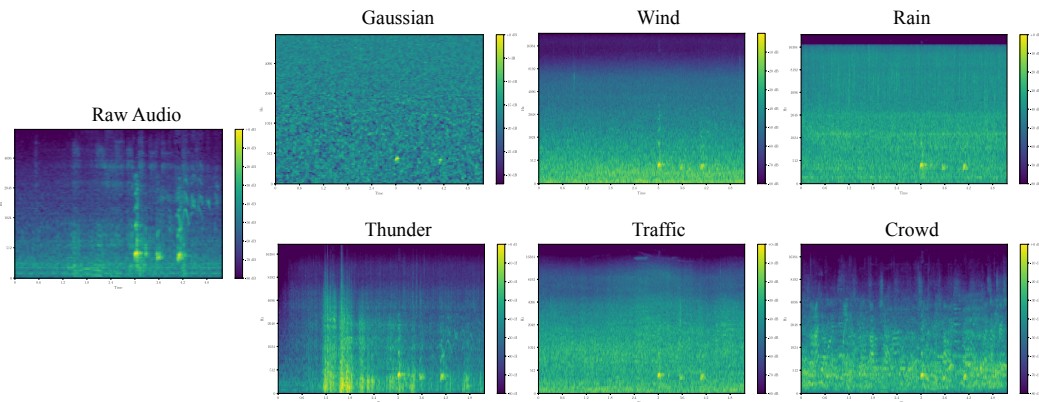

Figure 10: Mel spectrogram visualization of the raw audio and the corresponding audio corruption types on the constructed Kinetics-100-C benchmark.

contains $3,381$ video clips and we use the provided *video*, *optical flow*, and *audio* modalities for our experiments.

**Kinetics-600 (Carreira et al., 2018).** Kinetics-600 is a large-scale action recognition dataset comprising approximately $480k$ videos across 600 action categories. Each video is a 10-second clip of action moment annotated from YouTube videos. In our benchmark, we carefully selected a subset of 100 action classes from Kinetics-600 to mitigate the potential category overlap with the HAC dataset, with each class comprising roughly 250 video clips, yielding a total of $24,981$ video clips. We use the provided *video* and *audio* modalities to create a new *Kinetics-100-C* corruption dataset (Fig. 9 and Fig. 10) following the ideas in Hendrycks & Dietterich (2019) and Yang et al. (2024). We apply six different types of corruptions (Gaussian, Defocus, Frost, Brightness, Pixelate, and JPEG) on videos and six others (Gaussian, Wind, Traffic, Thunder, Rain, and Crowd) on audios. We randomly combine them to generate 6 different corruption shifts and all our experiments are conducted under the most severe corruption level of $5$ ((Hendrycks & Dietterich, 2019)). For example, *Defocus (v) + Wind (a)* means we add defocus corruption on video and wind corruption on audio.

**nuScenes (Caesar et al., 2020).** nuScenes is a large-scale dataset designed for autonomous driving research. It includes data collected from autonomous vehicles equipped with a comprehensive sensor suite, including cameras, LiDAR, radar, GPS, and IMU. The dataset provides 360° coverage of urban environments in Boston and Singapore, offering diverse weather and traffic scenarios. It includes $1,000$ driving scenes with rich annotations, such as 3D bounding boxes for 23 object classes, enabling tasks like object detection, tracking, and scene understanding. The scenes are split into $28,130$ training frames and $6,019$ validation frames.

| | Defocus (v) + Wind (a) | | | | Frost (v) + Traffic (a) | | | | Brightness (v) + Thunder (a) | | | | Pixelate (v) + Rain (a) | | | |
|---|---|---|---|---|---|---|---|---|---|---|---|---|---|---|---|---|
| | Acc↑ | FPR95↓ | AUROC↑ | H-score↑ | Acc↑ | FPR95↓ | AUROC↑ | H-score↑ | Acc↑ | FPR95↓ | AUROC↑ | H-score↑ | Acc↑ | FPR95↓ | AUROC↑ | H-score↑ |
| Source | 73.82 | 67.74 | 75.04 | 51.84 | 40.58 | 86.34 | 56.37 | 25.95 | 80.89 | 59.13 | 79.03 | 60.63 | 79.00 | 66.42 | 73.76 | 53.58 |
| Tent | 73.42 | 89.66 | 60.89 | 23.67 | 56.47 | 94.50 | 52.03 | 13.71 | 79.21 | 87.71 | 61.44 | 27.21 | 78.34 | 87.63 | 61.20 | 27.29 |
| SAR | 73.24 | 89.16 | 66.78 | 24.82 | 57.32 | 94.97 | 55.06 | 12.80 | 78.58 | 88.66 | 64.74 | 25.78 | 78.71 | 88.89 | 64.57 | 25.38 |
| OSTTA | 73.47 | 75.24 | 72.09 | 44.20 | 59.53 | 91.39 | 61.28 | 20.10 | 78.34 | 78.34 | 73.70 | 41.38 | 77.39 | 66.05 | 78.25 | 54.39 |
| UniEnt | 73.63 | 57.37 | 84.54 | 61.39 | 56.87 | 79.47 | 70.69 | 37.30 | 78.76 | 45.11 | 88.08 | 70.98 | 78.16 | 54.37 | 84.42 | 64.44 |
| READ | 74.18 | 48.32 | 85.06 | 67.28 | 60.92 | 64.61 | 76.78 | 52.00 | 79.08 | 40.18 | 88.31 | 73.74 | 78.18 | 41.87 | 88.02 | 72.54 |
| AEO (Ours) | 74.03 | 47.45 | 85.72 | 67.87 | 61.11 | 63.00 | 78.28 | 53.41 | 78.92 | 37.71 | 89.68 | 75.23 | 77.74 | 39.00 | 90.09 | 74.34 |
| | JPEG (v) + Crowd (a) | | | | Gaussian (v) + Gaussian (a) | | | | *Mean* | | | | | | | |
| | Acc↑ | FPR95↓ | AUROC↑ | H-score↑ | Acc↑ | FPR95↓ | AUROC↑ | H-score↑ | Acc↑ | FPR95↓ | AUROC↑ | H-score↑ | | | | |
| Source | 76.47 | 64.97 | 75.25 | 54.63 | 41.61 | 88.13 | 55.48 | 23.75 | 65.40 | 72.12 | 69.16 | 45.06 | | | | |
| Tent | 78.16 | 89.47 | 60.27 | 24.12 | 64.00 | 93.08 | 55.15 | 16.83 | 71.60 | 90.34 | 58.50 | 22.14 | | | | |
| SAR | 77.76 | 90.18 | 64.09 | 23.02 | 64.89 | 93.84 | 59.27 | 15.41 | 71.75 | 90.95 | 62.42 | 21.20 | | | | |
| OSTTA | 77.50 | 78.66 | 71.84 | 40.71 | 64.79 | 91.42 | 63.27 | 20.30 | 71.84 | 80.18 | 70.07 | 36.85 | | | | |
| UniEnt | 77.53 | 51.87 | 85.52 | 66.13 | 63.76 | 84.08 | 71.09 | 32.41 | 71.45 | 62.04 | 80.72 | 55.44 | | | | |
| READ | 77.95 | 43.47 | 86.72 | 71.34 | 66.08 | 55.82 | 81.82 | 60.01 | 72.73 | 49.05 | 84.45 | 66.15 | | | | |
| AEO (Ours) | 78.08 | 42.37 | 88.06 | 72.26 | 65.63 | 54.50 | 82.74 | 60.85 | 72.59 | 47.34 | 85.76 | **67.33** | | | | |

Table 12: Multimodal Open-set TTA with video and audio modalities on Kinetics-100-C, with corrupted video and audio modalities (severity level 3).

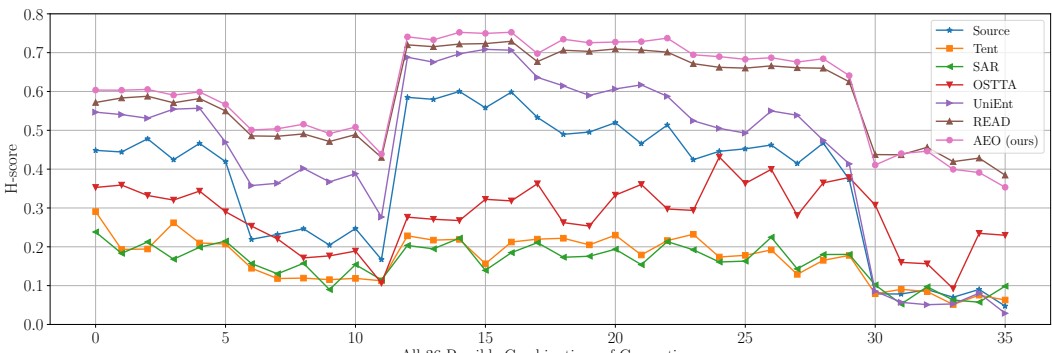

Figure 11: Detailed results on all 36 possible combinations of corruptions. Our AEO achieves the best on 31 of them.

### B.5 EXTENSION TO MORE MODALITIES

Our *AEO* framework is not limited to two modalities and can be easily extended to $M$ modalities. Given a sample x with $M$ modalities, we obtain prediction probabilities $\hat{p}$ from the combined embeddings of all modalities, and $\hat{p}^1$, $\hat{p}^2$, ..., $\hat{p}^M$ from each modality, all of which are of shape $[1, C]$, where $C$ represents the number of classes. In this case, $\mathcal{L}_{AdaEnt}$ is still in its original form as in Eq. (4) and $\mathcal{L}_{AdaEnt*}$ can be defined as:

$$\mathcal{L}_{AdaEnt*} = -\frac{1}{M} \sum_{i=1}^{M} H(\hat{p}^i) \cdot W_{ada}, \qquad (10)$$

and $\mathcal{L}_{AdaDis}$ becomes:

$$\mathcal{L}_{AdaDis} = -\frac{2}{M(M-1)} \sum_{i=1}^{M-1} \sum_{j=i+1}^{M} Dis(\hat{p}^i, \hat{p}^j) \cdot W_{ada}. \qquad (11)$$

The final loss is also the same as in Eq. (8). As shown in Tab. 2 and Tab. 23, our framework demonstrates strong performances when video, audio, and optical flow are all available.

## C ADDITIONAL EXPERIMENTAL RESULTS

### C.1 MORE RESULTS ON KINETICS-100-C DATASET WITH DIFFERENT SEVERITY LEVELS

In previous experiments, we used the most challenging setup (severity level 5) for Kinetics-100-C. In this section, we evaluate the performance of various methods under milder corruptions (severity level 3). As shown in Tab. 12, our AEO consistently outperforms baselines and achieves the best results at severity level 3, consistent with the findings at severity level 5.

| | Source | Tent | SAR | OSTTA | UniEnt | READ | AEO (ours) |
|---|---|---|---|---|---|---|---|
| H-score | 37.30 | 16.90 | 16.11 | 28.14 | 45.38 | 59.11 | **60.56** |

Table 13: Average H-score on all 36 possible combinations of corruptions.

| | Kinetics-100-C | | | |
|---|---|---|---|---|
| | Acc↑ | FPR95↓ | AUROC↑ | H-score↑ |
| Source | 61.06 | 73.09 | 68.82 | 43.51 |
| UniEnt | 66.87 | 59.55 | 80.31 | 56.37 |
| READ | 68.65 | 52.52 | 82.31 | 62.44 |
| AEO (Ours) | 68.39 | 49.85 | 84.70 | **64.39** |

Table 14: Multimodal Open-set TTA with video and audio modalities on Kinetics-100-C, with corrupted video and audio modalities (severity level 5). The corruptions for open-set data (HAC dataset) are mixed.

## C.2 EVALUATION RESULTS ON ALL 36 POSSIBLE COMBINATIONS OF CORRUPTIONS

In the previous experiments, we applied six types of corruption separately to video and audio, then randomly combined them to generate six distinct corruption shifts (e.g., Defocus (v) + Wind (a), Frost (v) + Traffic (a), Brightness (v) + Thunder (a), etc.). In total, there are 36 possible combinations. In this section, we comprehensively evaluate various baselines across all 36 combinations and present the results in Fig. 11 and Tab. 13. Our AEO achieves the best performance on 31 out of 36 combinations and obtains the highest average H-score, demonstrating its robustness under diverse corruption scenarios.

## C.3 ROBUSTNESS UNDER MIXED CORRUPTIONS FOR OPEN-SET DATA

By default, the corruptions applied to the open-set data (HAC dataset) are the same as those applied to the closed-set data (Kinetics-100) for each configuration on Kinetics-100-C. For instance, in the Defocus (v) + Wind (a) configuration, Defocus corruption is applied to video and Wind corruption to audio for both the HAC and Kinetics-100 datasets. In this section, we evaluate a more challenging setup, where the corruptions for the open-set data are mixed and randomly sampled from all six possible corruptions. For example, in the Defocus (v) + Wind (a) configuration, Defocus corruption is applied to video and Wind corruption to audio for Kinetics-100, while one of the six possible corruptions is randomly assigned to each sample in the HAC dataset. As shown in Tab. 14, our AEO consistently outperforms the baselines in this challenging setup.

## C.4 ROBUSTNESS UNDER DIFFERENT SCORE FUNCTIONS

The score function in Eq. (2) is flexible, with several options such as MSP (Hendrycks & Gimpel, 2017), MaxLogit (Hendrycks et al., 2022), Energy (Liu et al., 2020), and Entropy (Liu et al., 2023). In this paper, we use MSP as the default and evaluate other score functions on the HAC dataset using both video and audio, as shown in Tab. 15. Our AEO demonstrates low sensitivity to the choice of score functions, achieving comparable performance across different metrics.

## C.5 INFLUENCE OF $\mathcal{L}_{Div}$ TO THE PERFORMANCES

We investigate the impact of $\mathcal{L}_{Div}$ in Eq. (7) to the performances, a negative entropy loss term widely used in prior works (Zhou et al., 2023; Yang et al., 2024) to promote diversity in predictions. As shown in Tab. 16, removing $\mathcal{L}_{Div}$ results in performance on the EPIC-Kitchens dataset remaining comparable to the original, while significantly reducing performance on the HAC and Kinetics-100-C datasets. This demonstrates the critical role of $\mathcal{L}_{Div}$ in ensuring diversity in predictions.

## C.6 ABLATION ON EACH COMPONENT IN AMP

We analyze the contributions of each term in Adaptive Modality Prediction Discrepancy Optimization (AMP), specifically $\mathcal{L}_{AdaEnt*}$ and $\mathcal{L}_{AdaDis}$. The $\mathcal{L}_{AdaEnt*}$ term adaptively maximizes or minimizes the prediction entropy of each modality and $\mathcal{L}_{AdaDis}$ dynamically adjusts the prediction discrepancy between modalities based on whether the samples are known or unknown. As shown in Tab. 17, with

|  | MSP | MaxLogit | Energy | Entropy |
|---|---|---|---|---|
| Acc↑ | 59.53 | 59.53 | 59.53 | 59.53 |
| FPR95↓ | 66.75 | 65.25 | 65.02 | 66.14 |
| AUROC↑ | 72.50 | 73.07 | 73.31 | 73.52 |
| H-score↑ | 48.31 | 49.69 | 49.86 | 49.04 |

Table 15: Ablation on different score functions on HAC dataset.

|  | HAC | | | | EPIC-Kitchens | | | | Kinetics-100-C | | | |
|---|---|---|---|---|---|---|---|---|---|---|---|---|
|  | Acc↑ | FPR95↓ | AUROC↑ | H-score↑ | Acc↑ | FPR95↓ | AUROC↑ | H-score↑ | Acc↑ | FPR95↓ | AUROC↑ | H-score↑ |
| w/o $\mathcal{L}_{Div}$ | 59.54 | 67.74 | 73.05 | 46.62 | 48.79 | 45.78 | 86.11 | **56.40** | 61.47 | 66.26 | 78.25 | 50.15 |
| w/ $\mathcal{L}_{Div}$ | 59.53 | 66.75 | 72.50 | **48.31** | 50.93 | 48.65 | 85.58 | 56.18 | 63.88 | 54.13 | 82.22 | **60.07** |

Table 16: Ablation on $\mathcal{L}_{Div}$ to the final performances.

| $\mathcal{L}_{AdaDis}$ | $\mathcal{L}_{AdaEnt*}$ | HAC | | | |
|---|---|---|---|---|---|
|  |  | Acc↑ | FPR95↓ | AUROC↑ | H-score↑ |
|  |  | 59.76 | 67.58 | 71.53 | 47.79 |
| ✓ |  | 59.49 | 67.48 | 72.39 | 47.60 |
|  | ✓ | 59.85 | 67.41 | 72.36 | 47.74 |
| ✓ | ✓ | 59.53 | 66.75 | 72.50 | **48.31** |

Table 17: Ablation on each component in AMP on HAC dataset.

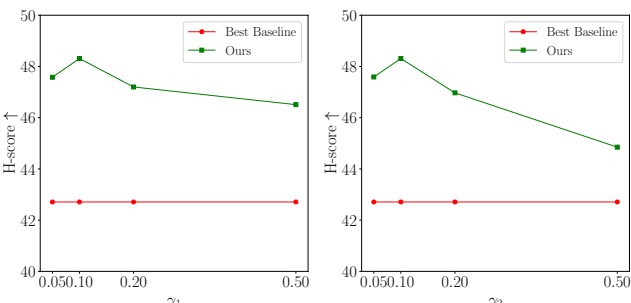

Figure 12: Sensitivity to loss hyperparameters $\gamma_1$ and $\gamma_2$.

only $\mathcal{L}_{AdaEnt*}$ or $\mathcal{L}_{AdaDis}$, the performance is very close to using $\mathcal{L}_{AdaEnt}$ alone. The best results are achieved when they are combined, which means $\mathcal{L}_{AdaEnt*}$ and $\mathcal{L}_{AdaDis}$ are complementary to each other.

### C.7 SENSITIVITY TO LOSS HYPERPARAMETERS

We evaluate the sensitivity of our method to the hyperparameters $\gamma_1$ and $\gamma_2$ in Eq. (8) by varying one hyperparameter at a time while keeping the others fixed. As shown in Fig. 12, our method consistently outperforms the best baseline, READ, across all parameter configurations. These results highlight the robustness of our approach and its low sensitivity to changes in hyperparameter settings.

### C.8 MORE VISUALIZATIONS ON PREDICTION SCORE DISTRIBUTIONS

Fig. 13 presents the prediction score distributions generated by various methods on the EPIC-Kitchens dataset before and after TTA. The score distributions of known and unknown samples exhibit significant overlap without TTA (Fig. 13 (a), (d), and (g)), leading to poor detection of unknown classes. Tent (Wang et al., 2021) minimizes the entropy of all samples, whether known or unknown, further compressing the score distributions (Fig. 13 (b), (e), and (h)) and making separation more difficult. In contrast, the score distributions produced by our method achieve better separation between known and unknown samples (Fig. 13 (c), (f), and (i)), thereby improving unknown class detection.

Fig. 14 depicts the model prediction entropy for known and unknown samples during online adaptation, evaluated batch by batch. Without TTA (Fig. 14 (a), (d), and (g)), the entropy values for known and unknown samples also show considerable overlap, resulting in weak unknown class detection. Tent (Wang et al., 2021) minimizes the entropy of all samples, further narrowing the gap between known and unknown samples (Fig. 14 (b), (e), and (h)). In contrast, the entropy values generated by

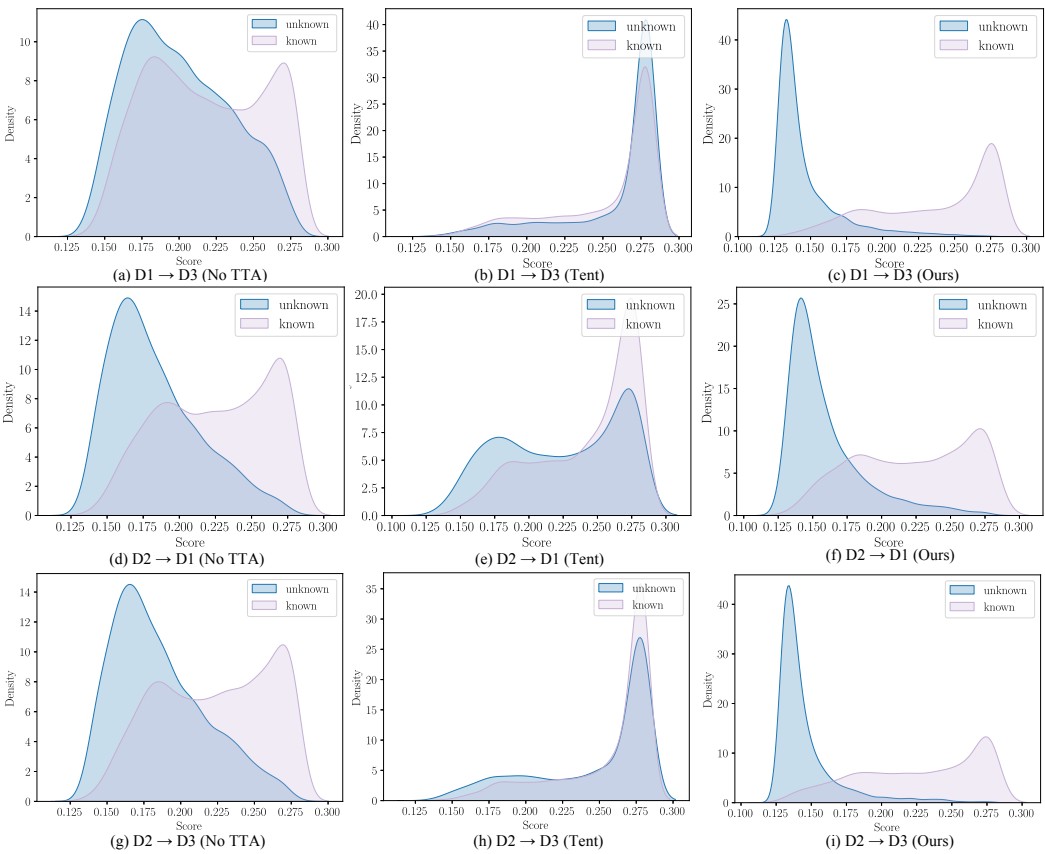

Figure 13: Prediction score distributions of different methods on the EPIC-Kitchens dataset before and after TTA.

our method enable better separation between known and unknown samples during online adaptation (Fig. 14 (c), (f), and (i)), thus enhancing the detection of unknown classes.

## C.9 DIFFERENT RANDOM SEEDS

We run experiments three times using different seeds on HAC dataset using video and audio (A → H adaptation) and then calculate the mean H-score to demonstrate the statistical significance of our methods. As illustrated in Fig. 15, training with *AEO* is statistically stable and consistently outperforms the baselines across various random seeds. In contrast, most baseline methods exhibit instability, with large variances across different runs.

## C.10 DETAILED RESULTS ON KINETICS-100-C DATASET IN CONTINUAL MM-OSTTA SETTING

Tab. 18 presents the detailed results on the Kinetics-100-C dataset under the *continual* MM-OSTTA setting, where the target domain distribution continually changes over time without resetting the model. We simulate continual TTA by repeating adaptation across continually changing domains (*Defocus (v) + Wind (a) → Frost (v) + Traffic (a) → Brightness (v) + Thunder (a) → ...*). Notably, our method achieves a $4.12\%$ increase in H-score after continual adaptation, indicating its robustness to error accumulation and its ability to continually optimize the entropy difference between known and unknown samples. Instead, most baseline methods suffer from severe performance degradation after continual adaptation.

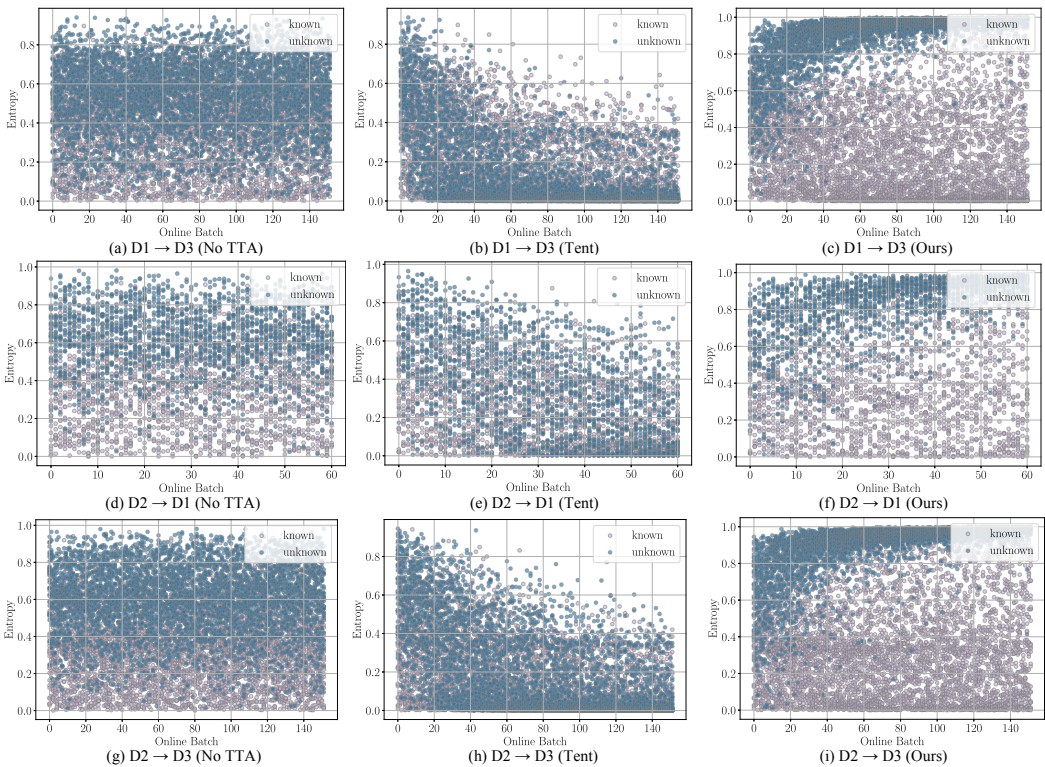

Figure 14: Model prediction entropy for known and unknown samples during online adaptation.

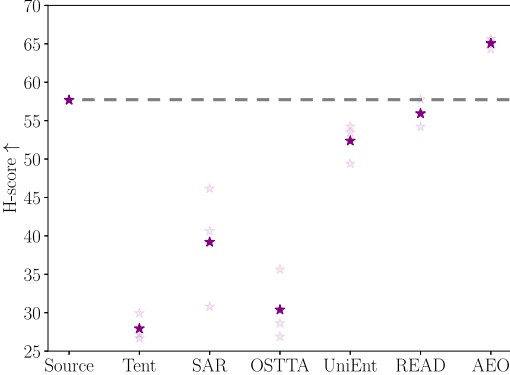

Figure 15: Experiments using three random seeds on HAC dataset using video and audio (A → H adaptation). Foreground points in bold show results averaged across three different seeds while background points, shown feint, indicate results from the underlying individual seeds.

## C.11 DETAILED RESULTS ON HAC DATASET IN CONTINUAL MM-OSTTA SETTING

Tab. 19 provides detailed results on the HAC dataset in the *continual* MM-OSTTA setting, where the target domain distribution continually changes over time and the model is never reset. We simulate three continual TTA scenarios (H → A → C, A → C → H, and C → H → A). Similar to the results on Kinetics-100-C, our method improves the H-score by 5.53% after continual adaptation, further demonstrating its robustness against error accumulation and its capability to constantly optimize the entropy difference between known and unknown samples. Meanwhile, most baseline methods show severe performance degradation after continual adaptation.

*t* ———→

| | Defocus (v) + Wind (a) | | | | Frost (v) + Traffic (a) | | | | Brightness (v) + Thunder (a) | | | | Pixelate (v) + Rain (a) | | | |
|---|---|---|---|---|---|---|---|---|---|---|---|---|---|---|---|---|
| | Acc↑ | FPR95↓ | AUROC↑ | H-score↑ | Acc↑ | FPR95↓ | AUROC↑ | H-score↑ | Acc↑ | FPR95↓ | AUROC↑ | H-score↑ | Acc↑ | FPR95↓ | AUROC↑ | H-score↑ |
| Source | 60.74 | 72.63 | 71.93 | 44.84 | 33.24 | 87.68 | 55.33 | 23.20 | 78.97 | 59.50 | 79.13 | 60.01 | 70.63 | 72.11 | 69.38 | 46.56 |
| Tent | 62.24 | 85.87 | 61.09 | 29.07 | 45.53 | 95.26 | 43.62 | 11.73 | 66.39 | 95.18 | 44.55 | 12.25 | 62.76 | 97.45 | 38.64 | 6.91 |
| SAR | 63.32 | 89.45 | 64.89 | 23.81 | 49.34 | 98.21 | 32.27 | 4.92 | 70.11 | 94.18 | 36.09 | 14.03 | 66.68 | 97.37 | 28.08 | 6.96 |
| OSTTA | 62.76 | 81.39 | 65.19 | 35.29 | 48.66 | 94.18 | 50.23 | 14.13 | 71.97 | 92.08 | 56.24 | 18.99 | 68.76 | 94.39 | 47.72 | 14.03 |
| UniEnt | 63.53 | 62.50 | 80.20 | 54.67 | 48.66 | 74.97 | 76.13 | 40.74 | 73.18 | 46.34 | 88.74 | 68.86 | 71.97 | 58.18 | 85.84 | 60.66 |
| READ | 64.24 | 59.08 | 80.19 | 57.17 | 54.84 | 64.00 | 78.98 | 51.13 | 76.76 | 33.92 | 91.76 | 76.81 | 73.05 | 29.97 | 92.80 | 77.43 |
| AEO (Ours) | 64.21 | 57.68 | 81.08 | **58.21** | 54.05 | 59.74 | 82.40 | **54.08** | 74.71 | 25.74 | 94.57 | **80.16** | 69.95 | 21.21 | 94.64 | **79.88** |

———————→

| | JPEG (v) + Crowd (a) | | | | Gaussian (v) + Gaussian (a) | | | | *Mean* | | | |
|---|---|---|---|---|---|---|---|---|---|---|---|---|
| | Acc↑ | FPR95↓ | AUROC↑ | H-score↑ | Acc↑ | FPR95↓ | AUROC↑ | H-score↑ | Acc↑ | FPR95↓ | AUROC↑ | H-score↑ |
| Source | 61.11 | 70.11 | 69.38 | 46.70 | 13.05 | 98.18 | 43.90 | 4.62 | 52.96 | 76.70 | 64.84 | 37.66 |
| Tent | 53.84 | 98.13 | 34.23 | 5.15 | 21.74 | 97.95 | 34.05 | 5.33 | 52.08 | 94.97 | 42.70 | 11.74 |
| SAR | 58.95 | 97.08 | 25.98 | 7.54 | 20.34 | 96.55 | 31.22 | 8.09 | 54.79 | 95.47 | 36.42 | 10.89 |
| OSTTA | 62.13 | 94.32 | 46.19 | 14.03 | 30.92 | 97.82 | 38.14 | 5.80 | 57.53 | 92.36 | 50.62 | 17.05 |
| UniEnt | 65.37 | 69.32 | 82.03 | 49.93 | 35.00 | 98.68 | 55.01 | 3.73 | 59.62 | 68.33 | 77.99 | 46.43 |
| READ | 66.84 | 34.53 | 90.82 | 72.73 | 30.89 | 68.21 | 72.41 | 38.64 | 61.10 | 48.29 | 84.49 | 62.32 |
| AEO (Ours) | 61.37 | 28.55 | 92.16 | **72.92** | 28.15 | 62.82 | 78.46 | **39.91** | 58.74 | 42.62 | 87.22 | **64.19** |

Table 18: **Continual** Multimodal Open-set TTA with video and audio modalities on Kinetics-100-C, with corrupted video and audio modalities (severity level 5).

*t* ———————————→   *t* ———————————→

| | H → A | | | | A → C | | | | A → C | | | | C → H | | | |
|---|---|---|---|---|---|---|---|---|---|---|---|---|---|---|---|---|
| | Acc↑ | FPR95↓ | AUROC↑ | H-score↑ | Acc↑ | FPR95↓ | AUROC↑ | H-score↑ | Acc↑ | FPR95↓ | AUROC↑ | H-score↑ | Acc↑ | FPR95↓ | AUROC↑ | H-score↑ |
| Source | 57.28 | 87.53 | 45.90 | 25.12 | 38.79 | 98.99 | 22.49 | 2.83 | 44.85 | 87.78 | 58.24 | 24.73 | 67.99 | 58.26 | 74.93 | 57.68 |
| Tent | 57.51 | 78.59 | 65.63 | 37.82 | 31.25 | 98.16 | 39.60 | 4.99 | 54.87 | 91.27 | 55.23 | 19.88 | 66.62 | 92.00 | 52.79 | 18.87 |
| SAR | 58.50 | 77.70 | 68.39 | 39.19 | 41.82 | 97.06 | 48.43 | 7.80 | 55.33 | 90.26 | 60.35 | 21.85 | 67.48 | 92.65 | 59.20 | 17.88 |
| OSTTA | 55.63 | 76.82 | 72.14 | 40.01 | 31.53 | 88.97 | 63.08 | 21.70 | 52.76 | 89.34 | 57.61 | 23.06 | 67.74 | 89.83 | 55.83 | 22.90 |
| UniEnt | 60.15 | 78.04 | 64.02 | 38.57 | 39.71 | 99.63 | 29.20 | 1.09 | 52.21 | 93.75 | 55.99 | 15.23 | 65.25 | 85.94 | 68.75 | 29.70 |
| READ | 57.84 | 68.87 | 69.58 | 47.03 | 42.65 | 87.78 | 54.99 | 24.30 | 54.14 | 84.10 | 58.05 | 30.43 | 68.93 | 61.21 | 71.49 | 55.27 |
| AEO (Ours) | 58.39 | 65.78 | 73.30 | **50.01** | 47.24 | 82.08 | 70.49 | **32.91** | 53.86 | 72.33 | 70.31 | **43.52** | 70.01 | 37.20 | 90.76 | **72.77** |

*t* ———————→

| | C → H | | | | H → A | | | | *Mean* | | | |
|---|---|---|---|---|---|---|---|---|---|---|---|---|
| | Acc↑ | FPR95↓ | AUROC↑ | H-score↑ | Acc↑ | FPR95↓ | AUROC↑ | H-score↑ | Acc↑ | FPR95↓ | AUROC↑ | H-score↑ |
| Source | 61.36 | 71.67 | 67.72 | 45.21 | 67.88 | 65.23 | 70.81 | 52.07 | 56.36 | 78.24 | 56.68 | 34.61 |
| Tent | 62.94 | 78.73 | 65.22 | 38.35 | 59.71 | 91.83 | 50.29 | 18.86 | 55.48 | 88.43 | 54.79 | 23.13 |
| SAR | 66.91 | 74.41 | 66.88 | 43.49 | 62.80 | 94.81 | 50.27 | 13.13 | 58.81 | 87.81 | 58.92 | 23.89 |
| OSTTA | 63.52 | 78.30 | 64.44 | 38.79 | 62.14 | 82.89 | 53.24 | 32.15 | 55.55 | 84.36 | 61.06 | 29.77 |
| UniEnt | 66.26 | 64.38 | 76.11 | 53.28 | 67.77 | 62.36 | 81.07 | 55.90 | 58.56 | 80.68 | 62.52 | 32.30 |
| READ | 61.07 | 69.79 | 69.55 | 46.98 | 61.92 | 69.21 | 66.31 | 47.09 | 57.76 | 73.49 | 65.00 | 41.85 |
| AEO (Ours) | 64.10 | 61.28 | 79.70 | **55.58** | 66.89 | 43.60 | 88.80 | **68.27** | 60.08 | 60.38 | 78.89 | **53.84** |

Table 19: **Continual** Multimodal Open-set TTA with video and audio modalities on HAC dataset, with corrupted video and audio modalities.

| | H → A | | | | H → C | | | | A → H | | | | A → C | | | |
|---|---|---|---|---|---|---|---|---|---|---|---|---|---|---|---|---|
| | Acc↑ | FPR95↓ | AUROC↑ | H-score↑ | Acc↑ | FPR95↓ | AUROC↑ | H-score↑ | Acc↑ | FPR95↓ | AUROC↑ | H-score↑ | Acc↑ | FPR95↓ | AUROC↑ | H-score↑ |
| Source | 57.28 | 87.53 | 45.90 | 25.12 | 38.79 | 98.99 | 22.49 | 2.83 | 67.99 | 58.26 | 74.93 | 57.68 | 44.85 | 87.78 | 58.24 | 24.73 |
| Tent | 57.51 | 78.59 | 65.63 | 37.82 | 41.64 | 94.03 | 47.25 | 14.11 | 70.58 | 87.82 | 61.63 | 26.67 | 54.87 | 91.27 | 55.23 | 19.88 |
| SAR | 58.50 | 77.70 | 68.39 | 39.19 | 46.32 | 95.04 | 48.67 | 12.31 | 70.22 | 72.39 | 68.88 | 46.17 | 55.33 | 90.26 | 60.35 | 21.85 |
| OSTTA | 55.63 | 76.82 | 72.14 | 40.01 | 44.49 | 89.98 | 58.36 | 21.52 | 71.09 | 86.59 | 61.74 | 28.62 | 52.76 | 89.34 | 57.61 | 23.06 |
| UniEnt | 60.15 | 78.04 | 64.02 | 38.57 | 40.99 | 98.71 | 31.86 | 3.61 | 68.85 | 64.10 | 72.81 | 53.46 | 52.21 | 93.75 | 55.99 | 15.23 |
| READ | 57.84 | 68.87 | 69.58 | 47.03 | 45.40 | 86.95 | 56.73 | 25.80 | 70.51 | 58.62 | 73.60 | 57.76 | 54.14 | 84.10 | 58.05 | 30.43 |
| AEO (Ours) | 57.84 | 67.11 | 72.16 | 48.74 | 45.59 | 89.71 | 57.02 | 21.95 | 70.44 | 47.95 | 81.33 | 65.64 | 53.86 | 72.33 | 70.31 | 43.52 |

| | C → H | | | | C → A | | | | *Mean* | | | |
|---|---|---|---|---|---|---|---|---|---|---|---|---|
| | Acc↑ | FPR95↓ | AUROC↑ | H-score↑ | Acc↑ | FPR95↓ | AUROC↑ | H-score↑ | Acc↑ | FPR95↓ | AUROC↑ | H-score↑ |
| Source | 61.36 | 71.67 | 67.72 | 45.21 | 67.88 | 65.23 | 70.81 | 52.07 | 56.36 | 78.24 | 56.68 | 34.61 |
| Tent | 62.94 | 78.73 | 65.22 | 38.35 | 66.78 | 80.68 | 63.56 | 36.38 | 59.05 | 85.19 | 59.75 | 28.87 |
| SAR | 66.91 | 74.41 | 66.88 | 43.49 | 66.67 | 63.91 | 70.11 | 52.66 | 60.66 | 78.95 | 63.88 | 35.94 |
| OSTTA | 63.52 | 78.30 | 64.44 | 38.79 | 65.56 | 73.73 | 64.03 | 43.52 | 58.84 | 82.46 | 63.05 | 32.58 |
| UniEnt | 66.26 | 64.38 | 76.11 | 53.28 | 67.55 | 54.53 | 79.28 | 60.72 | 59.34 | 75.59 | 63.35 | 37.48 |
| READ | 61.07 | 69.79 | 69.55 | 46.98 | 63.69 | 68.65 | 68.72 | 48.27 | 58.78 | 72.83 | 66.04 | 42.71 |
| AEO (Ours) | 64.10 | 61.28 | 79.70 | 55.58 | 65.34 | 62.14 | 74.46 | 54.40 | 59.53 | 66.75 | 72.50 | **48.31** |

Table 20: Multimodal Open-set TTA with video and audio modalities on HAC dataset.

## C.12 DETAILED RESULTS ON HAC DATASET USING DIFFERENT COMBINATION OF MODALITIES

Tab. 20 through Tab. 23 summarize the detailed results on the HAC dataset using different combinations of modalities. Most existing TTA methods struggle to generalize effectively in the challenging multimodal open-set TTA setup. While UniEnt (Gao et al., 2024) and READ (Yang et al., 2024) perform well and exceed the Source baseline, other TTA methods fail to achieve robust performance, underscoring the complexities of multimodal open-set adaptation. In contrast, our method demonstrates strong robustness across all modality combinations, significantly improving the Source baseline H-score by 13.70%, 12.12%, 9.12%, and 10.82% for the respective modality setups. These improvements highlight the effectiveness of our approach in handling diverse multimodal scenarios, even under challenging open-set conditions.

| | H → A | | | | H → C | | | | A → H | | | | A → C | | | |
|---|---|---|---|---|---|---|---|---|---|---|---|---|---|---|---|---|
| | Acc↑ | FPR95↓ | AUROC↑ | H-score↑ | Acc↑ | FPR95↓ | AUROC↑ | H-score↑ | Acc↑ | FPR95↓ | AUROC↑ | H-score↑ | Acc↑ | FPR95↓ | AUROC↑ | H-score↑ |
| Source | 63.13 | 85.43 | 50.36 | 28.75 | 44.03 | 98.90 | 25.55 | 3.09 | 72.39 | 52.70 | 77.53 | 62.69 | 48.44 | 84.19 | 52.72 | 29.16 |
| Tent | 62.69 | 78.37 | 65.36 | 38.72 | 42.19 | 96.05 | 46.36 | 10.05 | 69.14 | 85.94 | 60.09 | 29.35 | 49.72 | 94.03 | 46.67 | 14.35 |
| SAR | 63.58 | 75.06 | 69.16 | 42.68 | 46.78 | 95.40 | 50.39 | 11.60 | 70.87 | 87.53 | 62.11 | 27.17 | 52.02 | 96.69 | 44.88 | 8.73 |
| OSTTA | 62.25 | 65.56 | 75.25 | 51.38 | 44.03 | 91.36 | 58.60 | 19.29 | 66.76 | 90.99 | 59.61 | 21.02 | 50.92 | 96.69 | 44.97 | 8.72 |
| UniEnt | 64.24 | 84.33 | 62.54 | 31.46 | 45.40 | 99.08 | 33.38 | 2.63 | 70.66 | 38.57 | 88.95 | 71.99 | 53.22 | 77.57 | 62.07 | 37.74 |
| READ | 63.69 | 62.03 | 77.71 | 54.64 | 41.54 | 93.29 | 54.26 | 15.66 | 70.37 | 67.34 | 69.70 | 50.70 | 53.31 | 88.33 | 51.81 | 24.24 |
| AEO (Ours) | 64.02 | 56.95 | 81.81 | 58.74 | 46.32 | 85.57 | 64.39 | 28.19 | 68.93 | 48.67 | 86.21 | 65.81 | 52.30 | 83.55 | 62.14 | 31.25 |
| | C → H | | | | C → A | | | | Mean | | | | | | | |
| | Acc↑ | FPR95↓ | AUROC↑ | H-score↑ | Acc↑ | FPR95↓ | AUROC↑ | H-score↑ | Acc↑ | FPR95↓ | AUROC↑ | H-score↑ | | | | |
| Source | 57.82 | 67.56 | 70.46 | 48.14 | 61.04 | 65.34 | 72.28 | 50.79 | 57.80 | 75.69 | 58.15 | 37.11 | | | | |
| Tent | 66.33 | 82.84 | 64.21 | 33.74 | 61.59 | 83.00 | 63.97 | 33.08 | 58.61 | 86.71 | 57.78 | 26.55 | | | | |
| SAR | 64.96 | 82.91 | 65.34 | 33.63 | 62.80 | 77.92 | 66.34 | 39.33 | 60.17 | 85.92 | 59.70 | 27.19 | | | | |
| OSTTA | 66.11 | 89.55 | 63.13 | 23.69 | 60.60 | 84.55 | 64.16 | 30.99 | 58.45 | 86.45 | 60.95 | 25.85 | | | | |
| UniEnt | 61.36 | 66.83 | 75.03 | 50.19 | 65.67 | 62.69 | 74.15 | 54.04 | 60.09 | 71.51 | 66.02 | 41.34 | | | | |
| READ | 62.58 | 70.44 | 71.20 | 46.98 | 62.69 | 68.32 | 71.60 | 48.79 | 59.03 | 74.96 | 66.05 | 40.17 | | | | |
| AEO (Ours) | 61.93 | 58.04 | 80.45 | 57.24 | 62.80 | 61.70 | 74.73 | 54.14 | 59.38 | 65.65 | 74.96 | **49.23** | | | | |

Table 21: Multimodal Open-set TTA with video and optical flow modalities on HAC dataset.

| | H → A | | | | H → C | | | | A → H | | | | A → C | | | |
|---|---|---|---|---|---|---|---|---|---|---|---|---|---|---|---|---|
| | Acc↑ | FPR95↓ | AUROC↑ | H-score↑ | Acc↑ | FPR95↓ | AUROC↑ | H-score↑ | Acc↑ | FPR95↓ | AUROC↑ | H-score↑ | Acc↑ | FPR95↓ | AUROC↑ | H-score↑ |
| Source | 57.40 | 65.34 | 71.75 | 49.83 | 29.60 | 92.83 | 45.70 | 15.37 | 54.72 | 78.73 | 69.16 | 37.62 | 40.26 | 89.34 | 57.20 | 22.04 |
| Tent | 54.75 | 71.96 | 70.43 | 44.04 | 29.78 | 98.35 | 40.91 | 4.52 | 55.37 | 88.18 | 57.74 | 25.00 | 35.85 | 96.60 | 47.36 | 8.74 |
| SAR | 55.74 | 54.75 | 79.80 | 57.07 | 29.50 | 96.42 | 50.35 | 9.01 | 54.87 | 93.73 | 58.80 | 15.41 | 35.66 | 96.97 | 45.59 | 7.89 |
| OSTTA | 52.76 | 70.20 | 73.31 | 45.35 | 29.04 | 97.70 | 42.94 | 6.09 | 53.20 | 83.20 | 59.64 | 31.41 | 35.39 | 97.52 | 43.84 | 6.60 |
| UniEnt | 57.84 | 48.12 | 84.59 | 62.00 | 29.22 | 88.79 | 60.47 | 21.43 | 53.79 | 71.38 | 73.36 | 44.67 | 37.50 | 88.14 | 54.84 | 23.22 |
| READ | 54.53 | 61.15 | 77.81 | 52.70 | 29.96 | 88.97 | 61.09 | 21.37 | 53.42 | 72.53 | 71.41 | 43.40 | 36.86 | 86.58 | 52.07 | 24.82 |
| AEO (Ours) | 54.42 | 52.21 | 83.62 | 58.53 | 31.99 | 83.64 | 66.76 | 27.94 | 54.72 | 64.10 | 77.39 | 50.80 | 36.76 | 86.31 | 53.01 | 25.19 |
| | C → H | | | | C → A | | | | Mean | | | | | | | |
| | Acc↑ | FPR95↓ | AUROC↑ | H-score↑ | Acc↑ | FPR95↓ | AUROC↑ | H-score↑ | Acc↑ | FPR95↓ | AUROC↑ | H-score↑ | | | | |
| Source | 39.01 | 79.24 | 65.85 | 33.71 | 44.92 | 71.19 | 72.72 | 42.42 | 44.32 | 79.45 | 63.73 | 33.50 | | | | |
| Tent | 41.38 | 90.56 | 56.27 | 20.29 | 48.79 | 88.41 | 57.51 | 24.16 | 44.32 | 89.01 | 55.04 | 21.12 | | | | |
| SAR | 43.33 | 75.78 | 66.57 | 37.79 | 43.38 | 76.27 | 66.61 | 37.40 | 43.75 | 82.32 | 61.29 | 27.43 | | | | |
| OSTTA | 39.29 | 92.72 | 54.98 | 16.57 | 41.83 | 88.19 | 56.34 | 23.75 | 41.73 | 88.26 | 55.17 | 21.63 | | | | |
| UniEnt | 43.69 | 79.09 | 69.89 | 35.29 | 45.70 | 75.50 | 70.28 | 39.00 | 44.62 | 75.17 | 68.90 | 37.60 | | | | |
| READ | 39.37 | 76.42 | 67.89 | 36.35 | 46.91 | 75.28 | 66.55 | 39.06 | 43.51 | 76.82 | 66.14 | 36.28 | | | | |
| AEO (Ours) | 42.03 | 66.47 | 79.03 | 45.27 | 45.81 | 63.91 | 77.24 | 48.01 | 44.29 | 69.44 | 72.84 | **42.62** | | | | |

Table 22: Multimodal Open-set TTA with optical flow and audio modalities on HAC dataset.

| | H → A | | | | H → C | | | | A → H | | | | A → C | | | |
|---|---|---|---|---|---|---|---|---|---|---|---|---|---|---|---|---|
| | Acc↑ | FPR95↓ | AUROC↑ | H-score↑ | Acc↑ | FPR95↓ | AUROC↑ | H-score↑ | Acc↑ | FPR95↓ | AUROC↑ | H-score↑ | Acc↑ | FPR95↓ | AUROC↑ | H-score↑ |
| Source | 64.57 | 80.79 | 55.25 | 35.03 | 36.86 | 96.32 | 36.79 | 9.20 | 68.85 | 50.76 | 80.07 | 63.40 | 47.24 | 75.37 | 68.33 | 39.26 |
| Tent | 63.25 | 74.28 | 64.27 | 42.70 | 38.42 | 93.01 | 48.26 | 15.81 | 69.57 | 91.28 | 53.05 | 20.28 | 53.86 | 96.88 | 38.63 | 8.22 |
| SAR | 64.02 | 70.42 | 69.26 | 46.97 | 41.36 | 95.77 | 47.92 | 10.66 | 70.15 | 77.72 | 67.02 | 40.51 | 55.33 | 97.15 | 42.49 | 7.64 |
| OSTTA | 64.02 | 68.21 | 69.56 | 48.82 | 40.53 | 91.08 | 53.52 | 19.30 | 66.83 | 87.67 | 56.27 | 26.35 | 57.08 | 94.39 | 44.78 | 13.75 |
| UniEnt | 63.91 | 74.28 | 68.10 | 43.35 | 38.88 | 95.96 | 44.73 | 10.15 | 68.71 | 43.40 | 85.23 | 68.25 | 50.92 | 68.20 | 71.56 | 46.11 |
| READ | 63.91 | 61.26 | 76.13 | 54.95 | 42.74 | 87.22 | 59.05 | 25.30 | 66.33 | 60.99 | 73.51 | 55.23 | 51.38 | 83.92 | 60.28 | 30.54 |
| AEO (Ours) | 64.90 | 57.62 | 79.36 | 58.13 | 43.84 | 86.76 | 60.72 | 26.13 | 67.27 | 48.74 | 82.69 | 64.56 | 54.23 | 76.47 | 65.08 | 39.32 |
| | C → H | | | | C → A | | | | Mean | | | | | | | |
| | Acc↑ | FPR95↓ | AUROC↑ | H-score↑ | Acc↑ | FPR95↓ | AUROC↑ | H-score↑ | Acc↑ | FPR95↓ | AUROC↑ | H-score↑ | | | | |
| Source | 58.76 | 78.01 | 64.88 | 38.51 | 63.69 | 76.60 | 65.52 | 40.71 | 56.66 | 76.31 | 61.81 | 37.68 | | | | |
| Tent | 64.02 | 81.69 | 60.43 | 34.57 | 64.02 | 84.33 | 56.97 | 30.93 | 58.86 | 86.91 | 53.60 | 25.42 | | | | |
| SAR | 67.84 | 74.62 | 66.86 | 43.42 | 64.68 | 78.37 | 64.89 | 38.91 | 60.56 | 82.34 | 59.74 | 31.35 | | | | |
| OSTTA | 57.97 | 83.27 | 59.30 | 31.95 | 64.68 | 81.68 | 60.29 | 34.63 | 58.52 | 84.38 | 57.29 | 29.13 | | | | |
| UniEnt | 60.85 | 79.52 | 69.03 | 37.62 | 66.11 | 71.41 | 70.85 | 46.72 | 58.23 | 72.13 | 68.25 | 42.03 | | | | |
| READ | 60.85 | 76.86 | 68.50 | 40.41 | 59.60 | 75.28 | 66.21 | 41.47 | 57.47 | 74.26 | 67.28 | 41.32 | | | | |
| AEO (Ours) | 64.31 | 63.23 | 78.41 | 54.05 | 64.02 | 68.43 | 70.63 | 48.82 | 59.76 | 66.88 | 72.82 | **48.50** | | | | |

Table 23: Multimodal Open-set TTA with video, audio, and optical flow modalities on HAC dataset.

## C.13 FURTHER DISCUSSIONS ON THE NOVELTY OF THE PROPOSED FRAMEWORK

### C.13.1 COMPARISON WITH OOD DETECTION METHODS

The primary contribution of Liu et al. (2020) is the proposal of energy as an inference-time OOD score. While Liu et al. (2020) also uses energy as a learning objective, assigning lower energies to in-distribution (ID) data and higher energies to OOD data, it relies on auxiliary OOD training data. This assumption—knowing whether a sample is ID or OOD during training—does not hold in MM-OSTTA, where the task is to differentiate between ID and OOD samples without prior knowledge of their status. Therefore, the energy learning objective proposed in Liu et al. (2020) can't be used for MM-OSTTA directly. Additionally, Liu et al. (2020) assigns equal weights to all samples in the energy loss, which is impractical in MM-OSTTA. In MM-OSTTA, both distribution and label shifts are present, and the network often encounters samples with high uncertainty, making it unclear whether they are ID or OOD. Assigning equal weights to such samples can hinder adaptation performance.

To address the first challenge, we use entropy as a robust indicator of known and unknown samples to make an initial differentiation. Then we address the second challenge by assigning dynamic weights to samples based on their distance to an entropy threshold. This mitigates the impact of noisy or

|  | HAC | EPIC-Kitchens | Kinetics-100-C |
|---|---|---|---|
| Consistent Prediction | 47.61 | 55.12 | 58.91 |
| AEO (ours) | **48.31** | **56.18** | **60.07** |

Table 24: Ablation on consistent predictions across modalities to the final performances. The H-score is reported.

ambiguous samples, focusing optimization on reliable ones. These adaptations make our Unknown-aware Adaptive Entropy Optimization substantially different from the energy-based approach in Liu et al. (2020).

### C.13.2 COMPARISON WITH MULTIMODAL TTA METHODS

Both Shin et al. (2022) and Xiong et al. (2024) address the multimodal closed-set TTA by introducing a consistency loss between predictions of different modalities to make them close. However, this approach can be detrimental for open-set TTA, especially when dealing with unknown samples.

For known samples, enforcing consistent predictions across modalities helps the model make confident and accurate final predictions, as each modality supports the others. For unknown samples, enforcing consistent predictions across modalities can be detrimental. If both modalities have predictions with high entropy on unknown samples, enforcing consistent predictions across them has minimal impact. However, when the model suffers from overconfidence issues and outputs high prediction certainty (low entropy) for unknown class samples (Fig. 6 (a)), enforcing consistent predictions can make the model output final predictions with high certainty aligned with known classes, degrading the open-set performances.

To address this, we propose Adaptive Modality Prediction Discrepancy Optimization, which adaptively enforces consistency or discrepancy based on whether a sample likely belongs to a known or unknown class. This approach ensures that consistency is enforced only when beneficial, while discrepancy is maintained for unknown samples to prevent misalignment with known classes. Our ablation study highlights the necessity of this adaptive strategy. Enforcing consistent predictions across all modalities degrades performance across all datasets, as shown in Tab. 24.

### C.13.3 COMPARISON WITH OPEN-SET METHODS

Safaei et al. (2024) and Gao et al. (2024) also utilized the entropy difference between known and unknown samples in open-set settings. However, our work goes beyond merely leveraging this difference. In Sec. 3.1, we provide a deeper and more comprehensive analysis of this phenomenon. Specifically, we investigate the relationship between entropy differences and the performance of MM-OSTTA across various baselines, identifying potential failure modes that previous works did not address. Building on these insights, we propose the Adaptive Entropy-aware Optimization (AEO) framework, specifically designed for the MM-OSTTA setup to amplify the entropy difference between known and unknown samples dynamically during online adaptation.

### C.13.4 COMPARISON WITH SELECTIVE ENTROPY MINIMIZATION METHODS

Both SAR (Niu et al., 2023) and EATA (Niu et al., 2022) select reliable samples according to their entropy values for adaptation, excluding high-entropy samples from adaptation and assigning higher weights to test samples with lower prediction uncertainties. While they achieve better performance than Tent, they face two key limitations in the multimodal open-set TTA setting:

**Limited handling of unknown classes.** SAR and EATA focus solely on minimizing the entropy of samples with lower prediction uncertainties (likely belonging to known classes) while leaving the high-entropy samples (potentially unknown classes) unaddressed. This selective approach restricts their ability to amplify the entropy difference between known and unknown samples. In contrast, our proposed Adaptive Entropy-aware Optimization (AEO) dynamically optimizes the entropy of both known and unknown classes, resulting in a significantly larger entropy difference, as demonstrated in Fig. 2.

**Sensitivity to overlapping score distributions.** In the challenging open-set setting, the initial score distributions of known and unknown samples are often closely aligned and difficult to distinguish, as

shown in Fig. 6 (a). In such cases, SAR and EATA could potentially minimize the entropy of unknown samples incorrectly, leading to degraded performance that may even fall below the Source baseline. AEO addresses this issue by progressively increasing the entropy difference between known and unknown samples, mitigating the risk of negative adaptation and ensuring more reliable performance.

Additionally, we introduce Adaptive Modality Prediction Discrepancy Optimization, which exploits cross-modal interactions to further enhance the separation between known and unknown classes. These complementary approaches strengthen our framework's ability to handle open-set challenges more effectively than selective entropy minimization methods.

### C.13.5 PERFORMANCE IMPROVEMENT IN ACCURACY

While our AEO framework is indeed primarily designed to enhance unknown sample detection, it does not compromise accuracy (Acc) for closed-set adaptation. On the contrary, our results show that AEO achieves competitive accuracy across all datasets.

To clarify, the assertion that AEO sacrifices Acc for its advantages in unknown sample detection is not supported by our findings. For example, while READ excels on EPIC-Kitchens and Kinetics-100-C, it performs poorly on the HAC dataset. Conversely, SAR performs well on HAC dataset but poorly on EPIC-Kitchens and Kinetics-100-C datasets. Our AEO, however, consistently performs well across all datasets. To provide further clarity, we have computed the average Acc across all datasets (EPIC-Kitchens, HAC (video+audio, video+flow, flow+audio, video+audio+flow), and Kinetics-100-C). As shown in Tab. 25, our AEO achieves the highest overall performance in Acc, demonstrating its robustness.

Moreover, we have computed the average values of other metrics (FPR95, AUROC, and H-score) across all datasets. As shown in Tab. 26, AEO achieves the best overall performance across all metrics, with notable improvements in FPR95, AUROC, and H-score. While the improvement in Acc is relatively modest, it is significant that AEO balances both closed-set adaptation and unknown sample detection effectively.

In summary, our work is the first to tackle the challenging and practical problem of Multimodal Open-set Test-time Adaptation. AEO does not trade Acc for improved unknown sample detection but instead achieves a well-rounded performance across both closed-set and open-set adaptation objectives.

| | Source | Tent | SAR | OSTTA | UniEnt | READ | AEO (ours) |
|---|---|---|---|---|---|---|---|
| Acc↑ | 53.02 | 54.97 | 56.00 | 54.65 | 55.77 | 55.77 | **56.30** |

Table 25: Average accuracy (Acc) across all datasets.

| | Acc↑ | FPR95↓ | AUROC↑ | H-score↑ |
|---|---|---|---|---|
| Source | 53.02 | 77.58 | 61.86 | 35.78 |
| Tent | 54.97 | 88.84 | 55.40 | 22.68 |
| SAR | 56.00 | 85.36 | 60.94 | 26.71 |
| OSTTA | 54.65 | 86.40 | 59.24 | 26.23 |
| UniEnt | 55.77 | 71.51 | 70.22 | 41.62 |
| READ | 55.77 | 72.14 | 69.36 | 42.81 |
| AEO (Ours) | **56.30 (+0.30)** | **61.93 (+9.58)** | **76.82 (+6.60)** | **50.82 (+8.01)** |

Table 26: Average value of different metrics on all datasets.

