# OpenReview forum: "Towards Robust Multimodal Open-set Test-time Adaptation via Adaptive Entropy-aware Optimization"
_ICLR.cc/2025/Conference — ICLR 2025 Poster_

### Official Review · Reviewer_uXos · 2024-10-18

**Soundness:** 2
**Presentation:** 2
**Contribution:** 2
**Rating:** 6
**Confidence:** 4

**Summary:**

The paper presents a new problem setting for multimodal open-set test-time adaptation (MM-OSTTA), where the multimodal test data stream contains both known and unknown classes. The paper presents adaptive entropy-aware optimization (AEO) to tackle the MM-OSTTA problem. Intensive evaluations on various multimodal open-set scenarios demonstrate the effectiveness of AEO.

**Strengths:**

- Introduces a new problem setting of multimodal open-set TTA.
- The proposed method (AEO) performs robustly among various datasets and scenarios by successfully discriminating between known and unknown samples.

**Weaknesses:**

- The paper needs better motivation/discussion of the new setting (MM-OSTTA) regarding how/why existing open-set TTA approaches fail in multimodal settings.

- The method seems incremental, combining multiple loss terms of (1) entropy discrimination (similar to OOD detection works [a]), (2) entropy minimization (similar to TTA works), (3) discrepancy minimization across modalities, and (4) negative entropy loss (as cited in the paper).

- The design of AMP with three objectives requires further justification (e.g., deeper ablation study of AMP for each loss function) and an explanation of how they connect to contribute.


- The proposed method performs better with more unknown samples (Table 10), which is counter-intuitive and lacks proper explanation.




----

Citations
> [a] Liu, W., Wang, X., Owens, J. D., & Li, Y. (2020). Energy-based Out-of-distribution Detection. Advances in Neural Information Processing Systems, 2020-December. https://arxiv.org/abs/2010.03759v4

**Questions:**

- Please discuss how diverse open-set data (e.g., SoTTA [b]) would affect the methods and results.
- How is the method sensitive to loss hyperparameters $\gamma_1, \gamma_2$?
- Why did the authors select hyperparameter $\alpha=0.7$ where 0.8 outperforms in Figure 5?
- The paper used softmax probability as the prediction score, which does not align with the entropy-based adaptation method. Why did the authors decide to use softmax instead of entropy?


----

Citations
> [b] Gong, T., Kim, Y., Lee, T., Chottananurak, S., & Lee, S. J. (2024). SoTTA: Robust Test-Time Adaptation on Noisy Data Streams. Advances in Neural Information Processing Systems, 36.

---

> ### Author Response · Authors · 2024-11-18
> **The Response to Reviewer uXos (Part 1)**
>
> Thanks for your insightful reviews, and we appreciate your valuable suggestions! We provide the responses to your questions as follows:
>
> >**Q1**: The paper needs better motivation/discussion of the new setting (MM-OSTTA) regarding how/why existing open-set TTA approaches fail in multimodal settings.
>
> **A1**: Thanks for your suggestion! OSTTA [a] is the first work on open-set TTA and proposes to filter out samples whose confidence values are lower in the adapted model than in the original model, and then minimize the entropy of the remaining samples. However, there are two key issues with OSTTA. First, OSTTA assumes that confidence values for unknown samples are lower in the adapted model than in the original model. This assumption may not hold in multimodal open-set TTA settings, where complex interactions between modalities can cause the confidence of unknown samples to increase, leading to incorrect filtering. Second, OSTTA minimizes entropy only for selected samples while neglecting filtered samples. This limits its ability to fully exploit the entropy difference between known and unknown samples. As a result, OSTTA struggles to adapt effectively in multimodal open-set scenarios. As shown in Tables 1-3 of our paper, while OSTTA performs better than Tent, it consistently fails to outperform the Source baseline, highlighting its limited effectiveness in MM-OSTTA.
>
> UniEnt [b] distinguishes known and unknown samples based on the similarity between feature embeddings and source domain prototypes, applying entropy minimization to pseudo-known samples and entropy maximization to pseudo-unknown samples. However, UniEnt relies heavily on the quality of the embedding space to accurately detect unknown classes. In multimodal scenarios, where embedding spaces can be noisy or inconsistent across modalities, this dependence reduces its robustness.  Besides, UniEnt does not consider the potential benefits of leveraging interactions between modalities to improve open-set detection.
> In contrast, our method avoids dependence on embedding quality by directly using entropy as a robust indicator of known and unknown samples. Furthermore, we propose Adaptive Modality Prediction Discrepancy Optimization, which dynamically exploits modality interactions to amplify the entropy difference between known and unknown samples.
>
> Our proposed AEO framework addresses these limitations comprehensively, as evidenced by its performance across datasets. While UniEnt performs well in certain scenarios, it still shows a significant gap compared to our AEO. By directly tackling the shortcomings of existing approaches and leveraging multimodal interactions, AEO achieves superior performance in both closed-set and open-set adaptation, as shown in Tables 1–3 of the paper.
>
> [a] Lee et al. Towards Open-Set Test-Time Adaptation Utilizing the Wisdom of Crowds in Entropy Minimization. ICCV 2023
>
> [b] Gao et al. Unified Entropy Optimization for Open-Set Test-Time Adaptation. CVPR 2024

---

> ### Author Response · Authors · 2024-11-18
> **The Response to Reviewer uXos (Part 2)**
>
> >**Q2**: The method seems incremental, combining multiple loss terms of (1) entropy discrimination (similar to OOD detection works [a]), (2) entropy minimization (similar to TTA works), (3) discrepancy minimization across modalities, and (4) negative entropy loss (as cited in the paper).
>
> **A2**: Thank you for your comment! We would like to clarify that our method is not a simple combination of existing loss functions. Instead, we propose two novel loss components—**Unknown-aware Adaptive Entropy Optimization (UAE)** and **Adaptive Modality Prediction Discrepancy Optimization (AMP)**—designed specifically to address the unique challenges of **multimodal** open-set TTA.
>
> Existing works such as SAR [a] and EATA [b] select reliable samples for adaptation based on their entropy values, typically excluding high-entropy samples and assigning higher weights to low-entropy ones (samples likely belonging to known classes). While effective to some extent, these methods only minimize the entropy of low-uncertainty samples without addressing the unknown samples. This limitation reduces their ability to significantly amplify the entropy difference between known and unknown samples, which is critical for open-set detection.
>
> In contrast, our UAE loss adaptively optimizes the entropy of both known and unknown samples, dynamically increasing the entropy difference between them. As shown in Figure 2, this approach significantly enhances the separation between known and unknown samples, improving open-set detection performance.
>
> We further introduce the adaptive modality prediction discrepancy loss (AMP), which leverages the interaction between modalities to refine entropy optimization. AMP enforces consistent predictions across modalities for known samples while dynamically maximizing prediction discrepancy for unknown samples. This unique design enhances the entropy difference by effectively utilizing modality-specific information, a feature not addressed by existing TTA or open-set methods.
>
> The negative entropy loss used in our framework is a common practice in TTA to promote prediction diversity and prevent model collapse. While its inclusion aligns with prior works, it is only one component of our framework and not the primary contributor to its novelty or effectiveness.
>
> Thus, our framework is more than a combination of existing techniques. By addressing the specific limitations of prior methods and introducing UAE and AMP, our method significantly improves both open-set detection and closed-set adaptation, as demonstrated by our results.
>
> [a] Niu et al. Towards Stable Test-Time Adaptation in Dynamic Wild World. ICLR 2023
>
> [b] Niu et al. Efficient test-time model adaptation without forgetting. ICML 2022
> ___
> >**Q3**: The design of AMP with three objectives requires further justification (e.g., deeper ablation study of AMP for each loss function) and an explanation of how they connect to contribute.
>
> **A3**: Thanks for your insightful comment! To address your concern, we conducted a detailed ablation study to analyze the contributions of each component in AMP, specifically $L_{AdaEnt*}$ and $L_{AdaDis}$. The $L_{AdaEnt*}$ term adaptively maximizes or minimizes the prediction entropy of each modality. It ensures that known samples are confidently predicted while increasing the entropy for unknown samples, contributing to the separation of known and unknown classes. $L_{AdaDis}$ dynamically adjusts the prediction discrepancy between modalities, promoting consistent predictions for known samples and disagreement for unknown samples. This interaction leverages the complementary information across modalities to further enhance the entropy difference.
>
> As shown in Table 1 below, each loss term provides distinct contributions to the overall performance. Importantly, the best results are achieved when both components are combined, demonstrating their synergy and highlighting how they collectively enhance the adaptation process.
>
> | $L_{AdaDis}$ | $L_{AdaEnt*}$ | Acc↑  | FPR95↓ | AUROC↑ | H-score↑        |
> |-|-|-|-|-|-|
> | $\checkmark$|| 59.49 | 67.48  | 72.39  | 47.60     |
> | | $\checkmark$            | 59.85 | 67.41  | 72.36  | 47.74      |
> | $\checkmark$           | $\checkmark$            | 59.53 | 66.75  | 72.50  | **48.31**  |
>
> *Table 1: Ablation on each component in AMP.*
>
> $L_{Div}$ is a common practice in TTA to promote prediction diversity and prevent model collapse. As shown in Table 2 below (detailed results are in Table 16 in the revised paper), without $L_{Div}$, the performance on the EPIC-Kitchens dataset is close to one with $L_{Div}$, but the performances drop significantly for HAC and Kinetics-100-C datasets, verifying the important of $L_{Div}$ to ensure diversity in predictions.
>
> ||HAC|EPIC-Kitchens|Kinetics-100-C|
> |-|-|-|-|
> |w/o $L_{Div}$|46.62|**56.40**|50.15|
> |w/ $L_{Div}$|**48.31**|56.18|**60.07**|
>
> *Table 2: Ablation on $L_{Div}$ to the final performances. The H-score is reported.*

---

> ### Author Response · Authors · 2024-11-18
> **The Response to Reviewer uXos (Part 3)**
>
> >**Q4**: The proposed method performs better with more unknown samples (Table 10), which is counter-intuitive and lacks proper explanation.
>
> **A4**: Thanks for your insightful observation! When the ratio of unknown samples increases from 0.4 to 0.8, the fluctuation in the H-score remains within 5%, with a non-monotonic trend (initially decreasing at a ratio of 0.5 before increasing). This indicates that the relationship between the unknown sample ratio and the H-score is not straightforward but rather irregular, demonstrating that our AEO is robust across different ratios.
>
> Additionally, we observe that as the ratio of unknown samples increases:
>
> **1. Accuracy (Acc) decreases**: This is likely due to the increased presence of unknown samples, which introduces more noise and challenges the optimization process for the known classes.
>
> **2. FPR95 improves significantly**: With more unknown samples, our AEO is better able to optimize the entropy difference between known and unknown samples, thereby enhancing open-set detection performance.
>
> The improvement in FPR95 outweighs the reduction in accuracy, leading to an overall increasing trend in the H-score. This trend reflects the core strength of AEO in balancing closed-set accuracy and open-set detection, even under varying proportions of unknown samples.
>
> ___
> >**Q5**: Please discuss how diverse open-set data (e.g., SoTTA [a]) would affect the methods and results.
>
> **A5**: Thanks for your insightful suggestion! To explore the impact of diverse open-set data, we follow the challenging setup proposed in SoTTA and evaluate our method on Kinetics-100-C. In this setup, corruptions for open-set data are mixed and randomly sampled from all six possible corruptions. For example, for Defocus (v) + Wind (a), we add Defocus corruption on video and Wind corruption on audio for Kinetics-100, but we add one of the six possible corruptions randomly for each sample on the HAC dataset.
>
> As shown in Table 3 below, our AEO demonstrates strong performance under this more diverse and challenging open-set scenario, consistently outperforming baseline methods. This highlights the robustness of AEO in handling not only single-corruption scenarios but also more complex setups with diverse and mixed corruptions, further emphasizing its practical applicability in real-world settings with unpredictable noise and corruption patterns.
>
> |  | Acc↑  | FPR95↓ | AUROC↑ | H-score↑        |
> |-|-|-|-|-|
> |  Source|  61.06 | 73.09  |68.82  |43.51  |
> |    UniEnt|   66.87 | 59.55 | 80.31  |56.37    |
> |    READ | 68.65  |52.52  |82.31 | 62.44    |
> |    AEO (ours) |68.39  |49.85 | 84.70 | **64.39**|
>
> *Table 3: Multimodal Open-set TTA with video and audio modalities on Kinetics-100-C, with corrupted video and audio modalities  (severity level 5). The corruptions for open-set data (HAC dataset) are mixed. The average results are reported.*
>
>
> ___
> >**Q6**: How is the method sensitive to loss hyperparameters $\gamma_1$, $\gamma_2$?
>
> **A6**: Thanks for your insightful comment! We evaluated the sensitivity of our method to the hyperparameters $\gamma_1$ and $\gamma_2$ in Eq. (8) by varying one hyperparameter at a time while keeping the others fixed on the HAC dataset. Our findings, as illustrated in Table 4 and Table 5 below,  demonstrate that our method consistently outperforms the best baseline, READ, across all parameter settings. These results indicate that our approach is robust and less sensitive to variations in hyperparameter choices.
>
> |      $\gamma_1$  | 0.05 | 0.1  | 0.2   | 0.5 | Best Baseline |
> |-------------------------|--------|-------|-------|-------|--------|
> | **H-score**             | 47.58| 48.31 |47.20| 46.51| 42.71 |
>
> *Table 4: Ablation on $\gamma_1$.*
>
> |      $\gamma_2$  | 0.05 | 0.1  | 0.2   | 0.5 | Best Baseline |
> |-------------------------|--------|-------|-------|-------|--------|
> | **H-score**             | 47.59 |48.31 |46.97| 44.85| 42.71 |
>
> *Table 5: Ablation on $\gamma_2$.*

---

> ### Author Response · Authors · 2024-11-18
> **The Response to Reviewer uXos (Part 4)**
>
> >**Q7**: Why did the authors select hyperparameter where 0.8 outperforms in Figure 5?
>
> **A7**: Thanks for your insightful observation! In our experiments, we conducted hyperparameter ablations on the HAC dataset and observed that the performance of our method is relatively robust to different values of $\alpha$, with fluctuations of less than 1\% across 0.7, 0.8, and 0.9. While $\alpha = 0.8$ slightly outperforms the other values on HAC, we found that $\alpha = 0.7$ delivers better performance on the EPIC-Kitchens and Kinetics-100-C datasets.
>
> Given this observation, we set $\alpha = 0.7$ as the default value for consistency and generalizability across all datasets. This ensures that the hyperparameter selection is not overly tailored to one specific dataset, reflecting the broader applicability of our method.
> ___
>
> >**Q8**: The paper used softmax probability as the prediction score, which does not align with the entropy-based adaptation method. Why did the authors decide to use softmax instead of entropy?
>
> **A8**: Thank you for your insightful observation! The score function in Eq. (2) is indeed flexible, and multiple options such as MSP (Maximum Softmax Probability), MaxLogit, Energy, and Entropy can be used. In our paper, we use MSP as the default score function for simplicity.
>
> To further investigate the impact of different score functions, we conducted evaluations on the HAC dataset using video and audio, as detailed in Table 6 below. The results show that our AEO is robust to the choice of the score function, with performance remaining consistently strong across different metrics. This indicates that our method's effectiveness is not tied to a specific score function, further demonstrating its adaptability and generalizability.
>
> |    | MSP| MaxLogit  | Energy   | Entropy |
> |-------------------------|--------|-------|-------|-------|
> |  FPR95↓            |  66.75|  65.25 | 65.02|  66.14|
> |   AUROC↑ | 72.50|  73.07| 73.31 | 73.52|
>
> *Table 6: Ablation on different score functions on HAC dataset.*

---

> ### Comment · Reviewer_uXos · 2024-11-22
>
> Thank you for a thorough rebuttal. Here are a few suggestions I would like to make from the rebuttal:
> - Q1: Please include in the manuscript the discussion of how existing OSTTA fails in MM-OSTTA. For example, in the Introduction, instead of mentioning "none specifically address the challenge of MM-OSTTA," I suggest briefly mentioning the existing method's limitations.
> - Q3: An ablation study only with AdaEnt would be better for understanding.
> - Q7: To avoid confusion, please include hyperparameter sensitivity on multiple datasets in Figure 5.
> - Q8: Can you also report AUROC and H-Score?
>
> A few concerns remain:
> - Q2: I am still worried about the technical novelty - diverging the entropy between known and unknown samples is not a new concept.

---

> > ### Comment · Reviewer_uXos · 2024-11-22
> >
> > After reading others' rebuttals, I am also worried about the limited improvement in accuracy. For the score, I will wait for the other reviewers' responses.

---

> > > ### Author Response · Authors · 2024-11-25
> > > **Further clarification regarding performance improvement in terms of accuracy**
> > >
> > > Dear Reviewer uXos,
> > >
> > > Thank you once again for your insightful reviews. Here we would like to add further clarifications regarding your concerns on performance improvement in terms of accuracy.
> > >
> > > While our AEO framework is indeed primarily designed to enhance unknown sample detection, it **does not compromise accuracy (Acc)** for closed-set adaptation. On the contrary, our results show that AEO achieves competitive accuracy across all datasets.
> > >
> > > To clarify, the assertion that AEO sacrifices Acc for its advantages in unknown sample detection is **not supported by our findings**. For example, while READ excels on EPIC-Kitchens and Kinetics-100-C, it performs poorly on the HAC dataset. Conversely, SAR performs well on HAC dataset but poorly on EPIC-Kitchens and Kinetics-100-C datasets. Our AEO, however, consistently performs well across all datasets. To provide further clarity, we have **computed the average Acc across all datasets** (EPIC-Kitchens, HAC (video+audio, video+flow, flow+audio, video+audio+flow), and Kinetics-100-C). As shown in Table 1 below, our **AEO achieves the highest overall performance in Acc**, demonstrating its robustness.
> > >
> > > || Source | Tent  | SAR   | OSTTA | UniEnt | READ  | AEO (ours) |
> > > |-------------------------|--------|-------|-------|-------|--------|-------|------------|
> > > | **Acc↑**             | 53.02| 54.97| 56.00 |54.65| 55.77 |55.77 | **56.30**  |
> > >
> > > *Table 1: Average accuracy (Acc) across all datasets.*
> > >
> > > Moreover, we have computed the average values of other metrics (FPR95, AUROC, and H-score) across all datasets. As shown in Table 2, **AEO achieves the best overall performance across all metrics**, with notable improvements in FPR95, AUROC, and H-score. While the improvement in Acc is relatively modest, it is significant that AEO balances both closed-set adaptation and unknown sample detection effectively.
> > >
> > > |  | Acc↑  | FPR95↓ | AUROC↑ | H-score↑        |
> > > |-------------------------|-------|--------|--------|------------------|
> > > |  Source|  53.02 | 77.58 | 61.86 | 35.78|
> > > |    Tent|54.97 | 88.84 | 55.40 | 22.68|
> > > |    SAR |56.00 | 85.36 | 60.94 | 26.71|
> > > |  OSTTA|54.65 | 86.40 | 59.24 | 26.23|
> > > |    UniEnt|55.77 | 71.51 | 70.22 | 41.62|
> > > |    READ | 55.77 | 72.14 | 69.36 | 42.81|
> > > |    AEO (ours) |**56.30 (+0.30)** | **61.93 (+9.58)** | **76.82 (+6.60)** | **50.82 (+8.01)**|
> > >
> > > *Table 2: Average value of different metrics on all datasets.*
> > >
> > > In summary, our work is the **first** to tackle the challenging and practical problem of Multimodal Open-set Test-time Adaptation. AEO does not trade Acc for improved unknown sample detection but instead **achieves a well-rounded performance across both closed-set and open-set adaptation objectives**.
> > >
> > >
> > > We sincerely hope these updates and clarifications address your concerns. Your valuable feedback has significantly strengthened our work, and we are deeply grateful for the time and effort you have dedicated to reviewing our paper. Please let us know if there are any remaining questions or concerns we can address further. Thank you once again for your thoughtful comments!

---

> > > > ### Comment · Reviewer_uXos · 2024-11-26
> > > >
> > > > I appreciate the authors' further clarifications regarding the novelty and performance. Concerns about the performance have been addressed for me.
> > > >
> > > > Below is my original comment:
> > > > > The method seems incremental, combining multiple loss terms of (1) entropy discrimination (similar to OOD detection works [a]), (2) entropy minimization (similar to TTA works), (3) discrepancy minimization across modalities, and (4) negative entropy loss (as cited in the paper).
> > > >
> > > > I acknowledge that the current method is new in terms of TTA, as the authors claimed. However, entropy discrimination is similar to OOD detection works (as cited above). Discrepancy minimization across modalities can be easily found as a consistency loss in multimodal TTA works [c, d]. Therefore, the method is still incremental as a summation of multiple existing approaches.
> > > >
> > > > I would appreciate any further discussions regarding this.
> > > >
> > > >
> > > > [c] Shin, Inkyu, et al. "Mm-tta: multi-modal test-time adaptation for 3d semantic segmentation." Proceedings of the IEEE/CVF Conference on Computer Vision and Pattern Recognition. 2022.
> > > >
> > > > [d] Xiong, Baochen, et al. "Modality-Collaborative Test-Time Adaptation for Action Recognition." Proceedings of the IEEE/CVF Conference on Computer Vision and Pattern Recognition. 2024.

---

> > > > > ### Author Response · Authors · 2024-11-26
> > > > >
> > > > > Dear Reviewer uXos,
> > > > >
> > > > > We are delighted to hear that your concerns about the performance have been resolved, and we sincerely appreciate your continued engagement and insightful feedback. Below, we address your remaining concern regarding the novelty of our method in detail.
> > > > >
> > > > > >**Comparison with [a]**
> > > > >
> > > > > The primary contribution of [a] is the proposal of energy as an inference-time OOD score. While [a] also uses energy as a learning objective, assigning lower energies to in-distribution (ID) data and higher energies to OOD data, it relies on auxiliary OOD training data. This assumption—knowing whether a sample is ID or OOD during training—does not hold in MM-OSTTA, where the task is to differentiate between ID and OOD samples without prior knowledge of their status. Therefore, **the energy learning objective proposed in [a] can't be used for MM-OSTTA directly**. Additionally, [a] assigns equal weights to all samples in the energy loss, which is impractical in MM-OSTTA. In MM-OSTTA, both distribution and label shifts are present, and the network often encounters samples with high uncertainty, making it unclear whether they are ID or OOD. **Assigning equal weights to such samples can hinder adaptation performance**.
> > > > >
> > > > > To address the first challenge, **we use entropy as a robust indicator** of known and unknown samples to make an initial differentiation. Then we address the second challenge by **assigning dynamic weights to samples based on their distance to an entropy threshold**. This mitigates the impact of noisy or ambiguous samples, focusing optimization on reliable ones. These adaptations make our Unknown-aware Adaptive Entropy Optimization **substantially different** from the energy-based approach in [a].
> > > > >
> > > > >
> > > > > >**Comparison with [c] and [d]**
> > > > >
> > > > > Both [c] and [d] address the multimodal **closed-set** TTA by introducing a consistency loss between predictions of different modalities to make them close. However, this approach can be **detrimental for open-set TTA**, especially when dealing with unknown samples.
> > > > >
> > > > > For **known samples**, enforcing consistent predictions across modalities helps the model make confident and accurate final predictions, as each modality supports the others.
> > > > >
> > > > > For **unknown samples**, enforcing consistent predictions across modalities can be detrimental. If both modalities have predictions with high entropy on unknown samples, enforcing consistent predictions across them has minimal impact. However, when the model suffers from overconfidence issues and outputs high prediction certainty (low entropy) for unknown class samples (Figure 6 (a)), **enforcing consistent predictions can make the model output final predictions with high certainty aligned with known classes, degrading the open-set performances**.
> > > > >
> > > > > To address this, we propose Adaptive Modality Prediction Discrepancy Optimization, which adaptively enforces consistency or discrepancy based on whether a sample likely belongs to a known or unknown class. This approach **ensures that consistency is enforced only when beneficial, while discrepancy is maintained for unknown samples to prevent misalignment with known classes**. Our ablation study highlights the necessity of this adaptive strategy. Enforcing consistent predictions across modalities for all samples degrades performance across all datasets, as shown in Table 1.
> > > > >
> > > > > |                 | HAC | EPIC-Kitchens | Kinetics-100-C |
> > > > > |-----------------|-----|---------------|----------------|
> > > > > | Consistent Prediction |  47.61   |        55.12       |          58.91      |
> > > > > | AEO (ours)      |    **48.31** |       **56.18**        |      **60.07**          |
> > > > >
> > > > > *Table 1: Ablation on consistent predictions across modalities to the final performances. The H-score is reported.*
> > > > >
> > > > > >**Summary and Additional Discussions**
> > > > >
> > > > > **We have included detailed discussions comparing our method with [a], [c], and [d] in Section C.14 of the revised paper.** Our proposed method is novel in its ability to handle the unique challenges of Multimodal Open-set Test-time Adaptation, particularly the **adaptive handling of known and unknown samples**.
> > > > >
> > > > > We sincerely hope these updates and clarifications address your concerns. Your valuable feedback has significantly strengthened our work, and we are deeply grateful for the time and effort you have dedicated to reviewing our paper. Please let us know if there are any remaining questions or concerns we can address further. Thank you once again for your thoughtful comments!

---

> > > > > > ### Comment · Reviewer_uXos · 2024-11-27
> > > > > >
> > > > > > Thank you for the detailed response. I will raise the score.
> > > > > >
> > > > > > Please consider revising the main manuscript to emphasize the importance of the adaptive weight.
> > > > > >
> > > > > > + Minor: Typo in Line 222: "$W_{ada}$ is *he* adaptive ..."

---

> > > > > > > ### Author Response · Authors · 2024-11-27
> > > > > > > **Thanks for raising the score!**
> > > > > > >
> > > > > > > Dear Reviewer uXos,
> > > > > > >
> > > > > > > We are glad to hear that we have addressed your concerns and that you raised your score to 6! We have fixed the typo and added Line 203 to refer to the discussion on the adaptive weight in Appendix C.14. Thanks for spending a significant amount of time on our submission and giving lots of valuable and insightful suggestions, which make our paper even stronger!

---

> > ### Author Response · Authors · 2024-11-23
> >
> > Thank you for your thorough follow-up comments and valuable suggestions! We have carefully updated the manuscript to incorporate your feedback and address your concerns.
> >
> > For Q1, we add lines 72-75 to briefly discuss the existing method's limitations. For Q3, we update Table 17, where the first row is the result with only $L_{AdaEnt}$. For Q7, we add Figures 13 and 14 for sensitivity analysis on EPIC-Kitchens and Kinetics-100-C respectively. For Q8, we add Acc and H-score values in Table 15.
> >
> > **Addressing Concerns on Novelty**
> >
> > We understand your concern that increasing the entropy difference between known and unknown samples is not a novel concept. However, **the core novelty of our work lies in how we achieve this effectively within the challenging MM-OSTTA setting**.
> >
> > Unlike existing methods, our UAE loss adaptively optimizes entropy based on the uncertainty of each sample. Samples with entropy values close to the threshold $\alpha$ in $W_{ada}$ (where the model is uncertain whether the sample is known or unknown) are given low weights close to 0, **reducing the negative influence of noisy or ambiguous samples**. Conversely, samples with very high or very low entropy (those most likely to be unknown or known, respectively) are assigned higher weights, focusing optimization on these critical cases. This adaptive weighting mechanism is **unique to our framework** and significantly enhances the separation between known and unknown samples.
> >
> > Besides, our AMP **exploits modality interactions** to further amplify entropy differences between known and unknown samples, which is **not addressed by existing TTA or open-set methods**. UAE and AMP work synergistically to tackle the multimodal and open-set challenges in MM-OSTTA and these innovations allow our method to achieve state-of-the-art performance in MM-OSTTA, as demonstrated by our extensive experiments.
> >
> > We sincerely hope these updates and clarifications address your concerns. Thank you again for your valuable comments, which have helped strengthen and refine our paper!

---

### Official Review · Reviewer_SHVx · 2024-10-27

**Soundness:** 3
**Presentation:** 3
**Contribution:** 2
**Rating:** 6
**Confidence:** 4

**Summary:**

In this paper, the authors present Adaptive Entropy-aware Optimization (AEO) to tackle Multimodal Open-set Test-time Adaptation (MM-OSTTA), which consists of Unknown-aware Adaptive Entropy Optimization (UAE) and Adaptive Modality Prediction Discrepancy Optimization (AMP). To thoroughly evaluate the proposed methods in the MMOSTTA setting, the authors establish a new benchmark derived from existing datasets.

**Strengths:**

1.The proposed method is easy to understand and the source code and benchmarks will be made publicly available.

2.A new benchmark that incorporates five modalities: video, audio, and optical flow for action recognition, as well as LiDAR and camera for 3D semantic segmentation is proposed.

3.The analysis on the entropy difference between known and unknown samples is reasonable.

**Weaknesses:**

1.It seems that the proposed Unknown-aware Adaptive Entropy Optimization has already been used in some of the existing works of wild TTA.

2.In the experimental settings, the authors say that all experiments are conducted under the most severe corruption level 5. However, it may be unrealistic in real-world applications. What about the performance under more lower corruption level?

3.More recent methods should be included for comparisons in Table 1 and 2, i.e., [1-2]

4.It seems that the performance of the proposed method on Acc is generally inferior to previous methods in Table 1 and 2.


[1]Two-Level Test-Time Adaptation in Multimodal Learning

[2]COME: Test-time adaption by Conservatively Minimizing Entropy

There is one minor issue that will not affect the rating:

1.It would be better for demonstration if the second row in Table 1 fills the entire page

**Questions:**

What is the underlying reason for the performance gap between the proposed method and previous ones on Acc metric?

---

> ### Author Response · Authors · 2024-11-18
> **The Response to Reviewer SHVx (Part 1)**
>
> Thanks for your insightful reviews, and we appreciate your valuable suggestions! We provide the responses to your questions as follows:
>
> >**Q1**: It seems that the proposed Unknown-aware Adaptive Entropy Optimization has already been used in some of the existing works of wild TTA.
>
> **A1**: Thanks for your insightful comment! Some existing approaches like SAR [a] and EATA [b] use partly similar concepts. Both SAR and EATA select reliable samples according to their entropy values for adaptation, excluding high-entropy samples from adaptation and assigning higher weights to test samples with lower prediction uncertainties. While SAR achieves better performance than Tent, it faces two key limitations in the multimodal open-set TTA setting:
>
> **1. Limited handling of unknown classes.**
> SAR focuses solely on minimizing the entropy of samples with lower prediction uncertainties (likely belonging to known classes) while leaving the high-entropy samples (potentially unknown classes) unaddressed. This selective approach restricts its ability to amplify the entropy difference between known and unknown samples. In contrast, our proposed Adaptive Entropy-aware Optimization (AEO) dynamically optimizes the entropy of both known and unknown classes, resulting in a significantly larger entropy difference, as demonstrated in Figure 2.
>
> **2. Sensitivity to overlapping score distributions.**
> In the challenging open-set setting, the initial score distributions of known and unknown samples are often closely aligned and difficult to distinguish, as shown in Figure 6 (a). In such cases, SAR could potentially minimize the entropy of unknown samples incorrectly, leading to degraded performance that may even fall below the Source baseline. AEO addresses this issue by progressively increasing the entropy difference between known and unknown samples, mitigating the risk of negative adaptation and ensuring more reliable performance.
>
> Additionally, we introduce **Adaptive Modality Prediction Discrepancy Optimization**, which exploits cross-modal interactions to further enhance the separation between known and unknown classes. These complementary approaches strengthen our framework's ability to handle open-set challenges more effectively than selective entropy minimization methods.
>
> [a] Niu et al. Towards Stable Test-Time Adaptation in Dynamic Wild World. ICLR 2023
>
> [b] Niu et al. Efficient test-time model adaptation without forgetting. ICML 2022
> ___
>
> >**Q2**: In the experimental settings, the authors say that all experiments are conducted under the most severe corruption level 5. However, it may be unrealistic in real-world applications. What about the performance under more lower corruption level?
>
> **A2**: Thanks for your insightful suggestion! To address your concern, we have conducted additional experiments under a milder corruption level (severity level 3) to evaluate the performance of different methods in more realistic scenarios. As shown in Table 1 below (detailed results are in Table 12 in the revised paper), our **AEO consistently outperforms all baselines** and achieves the best performance at severity level 3, aligning with the findings at severity level 5.
>
> |  | Acc↑  | FPR95↓ | AUROC↑ | H-score↑        |
> |-------------------------|-------|--------|--------|------------------|
> |  Source|  65.40 |72.12 |69.16 |45.06     |
> |    Tent|  71.60  |90.34  |58.50 | 22.14    |
> |    SAR | 71.75  |90.95  |62.42  |21.20  |
> |  OSTTA| 71.84 | 80.18 | 70.07 | 36.85  |
> |    UniEnt|   71.45 | 62.04 | 80.72  |55.44     |
> |    READ | 72.73  |49.05 | 84.45 | 66.15    |
> |    AEO (ours) |  72.59  |47.34 | 85.76  |**67.33**      |
>
> *Table 1: Multimodal Open-set TTA with video and audio modalities on Kinetics-100-C, with corrupted video and audio modalities  (severity level 3). The average results are reported.*
> ___
>
> >**Q3**: More recent methods should be included for comparisons in Table 1 and 2, i.e., [1-2]
>
> **A3**: Thanks for your suggestion! We further compared our method to 2LTTA [1] and COME [2], as shown in Table 2 below (detailed results are in Table 18 in the revised paper). Both of them are designed for closed-set TTA and achieve limited performances in the challenging multimodal open-set TTA setting.
>
> |                 | HAC | EPIC-Kitchens |
> |-----------------|-----|---------------|
> |2LTTA|  32.39   |          21.27     |
> |COME|   31.61  |    21.91           |
> | AEO (Ours) |    **48.31** |      **56.18**        |
>
> *Table 2: Comparison with more recent methods. The H-score is reported.*

---

> ### Author Response · Authors · 2024-11-18
> **The Response to Reviewer SHVx (Part 2)**
>
> >**Q4**: It seems that the performance of the proposed method on Acc is generally inferior to previous methods in Table 1 and 2.
>
> **A4**: Thanks for your insightful comment! In multimodal open-set TTA, model adaptation serves a dual purpose: improving the accuracy of **known samples** (closed-set adaptation) and effectively detecting **unknown samples** (open-set adaptation), and there is typically a trade-off between them. Both objectives are critical, particularly for safety-critical applications like autonomous driving and fraud detection, where robust unknown sample detection is vital for reliability.
>
> For the closed-set adaptation ability to known samples, we observe a common phenomenon in existing multimodal TTA work [a], where the improvement in accuracy is low compared to the Source baseline. In [a], they report improvements of only 2.6% on Kinetics50-C and 0.9% on VGGSound-C compared to the Source baseline. As a comparison, in our MM-OSTTA setting, our method improves the accuracy of 0.92% on EPIC-Kitchens dataset, 3.17% on HAC dataset, and 10.92% on Kinetics-100-C compared to the Source baseline. Compared to other SOTA baselines, our AEO is consistently among the top performers, with differences in accuracy being less than 1% in most cases compared to the best baseline.
>
> Furthermore, we observed that Tent, which minimizes the entropy of all samples indiscriminately (whether known or unknown), also improves the closed-set accuracy of 3.0% on HAC dataset, and 8.89% on Kinetics-100-C compared to the Source baseline (with terrible open-set adaptation performances on AUROC and FPR95). This observation suggests that adapting improperly to unknown samples can sometimes improve closed-set accuracy, while substantially explaining why other baseline methods also achieve competitive results on closed-set accuracy although with poor open-set performances.
>
> For open-set adaptation ability to unknown samples, our **AEO demonstrates significant superiority** over other baselines, improving the FPR95 of 30.47% on EPIC-Kitchens dataset, 11.49% on HAC dataset, and 22.57% on Kinetics-100-C compared with the Source baseline, and 17.42%, 6.08%, 2.11% respectively compared with the best baseline.
>
> Our AEO uniquely balances both closed-set and open-set adaptation, achieving strong performance across both objectives. This capability is **crucial for real-world safety-critical applications**, where both accurate predictions on known classes and reliable identification of unknown samples are essential for system robustness and reliability.
>
> We hope this clarifies the significant advantages of AEO and its ability to meet the dual objectives of multimodal open-set TTA. Thank you for helping us highlight these key aspects of our work.
>
> [a] Yang et al, Test-time Adaptation against Multi-modal Reliability Bias, in ICLR 2024
> ___
>
> >**Q5**: It would be better for demonstration if the second row in Table 1 fills the entire page.
>
> **A5**: Thanks for your suggestion! We have removed the extra lines in the table to make it look better.

---

> ### Author Response · Authors · 2024-11-24
>
> Dear Reviewer SHVx, we’ve carefully addressed all your insightful comments and incorporated your suggestions in the revised manuscript. Your feedback has been invaluable in improving the quality of our work, and we greatly appreciate your thoughtful review.
>
> As the discussion period is nearing its end, I wanted to ensure you’ve had the opportunity to review our responses and updated paper. Please let us know if there are any remaining questions or concerns we can address further.
>
> Thank you once again for your time and consideration.

---

> > ### Author Response · Authors · 2024-11-25
> > **Further clarification regarding performance improvement in terms of accuracy**
> >
> > Dear Reviewer SHVx,
> >
> > Thank you once again for your insightful reviews. Here we would like to add further clarifications regarding your concerns on performance improvement in terms of accuracy.
> >
> > While our AEO framework is indeed primarily designed to enhance unknown sample detection, it **does not compromise accuracy (Acc)** for closed-set adaptation. On the contrary, our results show that AEO achieves competitive accuracy across all datasets.
> >
> > To clarify, the assertion that AEO sacrifices Acc for its advantages in unknown sample detection is **not supported by our findings**. For example, while READ excels on EPIC-Kitchens and Kinetics-100-C, it performs poorly on the HAC dataset. Conversely, SAR performs well on HAC dataset but poorly on EPIC-Kitchens and Kinetics-100-C datasets. Our AEO, however, consistently performs well across all datasets. To provide further clarity, we have **computed the average Acc across all datasets** (EPIC-Kitchens, HAC (video+audio, video+flow, flow+audio, video+audio+flow), and Kinetics-100-C). As shown in Table 1 below, our **AEO achieves the highest overall performance in Acc**, demonstrating its robustness.
> >
> > || Source | Tent  | SAR   | OSTTA | UniEnt | READ  | AEO (ours) |
> > |-------------------------|--------|-------|-------|-------|--------|-------|------------|
> > | **Acc↑**             | 53.02| 54.97| 56.00 |54.65| 55.77 |55.77 | **56.30**  |
> >
> > *Table 1: Average accuracy (Acc) across all datasets.*
> >
> > Moreover, we have computed the average values of other metrics (FPR95, AUROC, and H-score) across all datasets. As shown in Table 2, **AEO achieves the best overall performance across all metrics**, with notable improvements in FPR95, AUROC, and H-score. While the improvement in Acc is relatively modest, it is significant that AEO balances both closed-set adaptation and unknown sample detection effectively.
> >
> > |  | Acc↑  | FPR95↓ | AUROC↑ | H-score↑        |
> > |-------------------------|-------|--------|--------|------------------|
> > |  Source|  53.02 | 77.58 | 61.86 | 35.78|
> > |    Tent|54.97 | 88.84 | 55.40 | 22.68|
> > |    SAR |56.00 | 85.36 | 60.94 | 26.71|
> > |  OSTTA|54.65 | 86.40 | 59.24 | 26.23|
> > |    UniEnt|55.77 | 71.51 | 70.22 | 41.62|
> > |    READ | 55.77 | 72.14 | 69.36 | 42.81|
> > |    AEO (ours) |**56.30 (+0.30)** | **61.93 (+9.58)** | **76.82 (+6.60)** | **50.82 (+8.01)**|
> >
> > *Table 2: Average value of different metrics on all datasets.*
> >
> > In summary, our work is the **first** to tackle the challenging and practical problem of Multimodal Open-set Test-time Adaptation. AEO does not trade Acc for improved unknown sample detection but instead **achieves a well-rounded performance across both closed-set and open-set adaptation objectives**.
> >
> >
> > We sincerely hope these updates and clarifications address your concerns. Your valuable feedback has significantly strengthened our work, and we are deeply grateful for the time and effort you have dedicated to reviewing our paper. As the discussion period is nearing its end, I wanted to ensure you’ve had the opportunity to review our responses and updated paper. Please let us know if there are any remaining questions or concerns we can address further. Thank you once again for your thoughtful comments!

---

> ### Author Response · Authors · 2024-11-27
>
> Dear Reviewer SHVx,
>
> We deeply appreciate your thoughtful feedback, which has helped us strengthen our work. Based on your insightful comments, we have incorporated additional ablations and analyses into the final version of the revised manuscript. Below is a summary of the updates:
>
> We have added Section **C.14.4 COMPARISON WITH SELECTIVE ENTROPY MINIMIZATION METHODS** to discuss the limitations of existing works of wild TTA (EATA [1] and SAR [2]) and highlight the unique advantages of our AEO.
>
> We have added Section **C.1 MORE RESULTS ON KINETICS-100-C DATASET WITH DIFFERENT SEVERITY LEVELS and Tab. 12** to include results on corruptions of severity level 3.
>
> We have added Section **C.8 COMPARISON WITH MORE RECENT METHODS and Tab. 18** to compare our method with 2LTTA [3] and COME [4].
>
> We have added Section **C.14.5 PERFORMANCE IMPROVEMENT IN ACCURACY and Tab. 26-27** to address your concerns regarding the Acc metric.
>
> [1] Niu et al., Efficient test-time model adaptation without forgetting. ICML'22.
>
> [2] Niu et al., Towards stable test-time adaptation in dynamic wild world. ICLR'23.
>
> [3] Lei et al., Two-Level Test-Time Adaptation in Multimodal Learning. ICML'24 Workshop
>
> [4] Zhang et al., COME: Test-time adaption by Conservatively Minimizing Entropy, arXiv'24
>
> We sincerely hope these updates and clarifications address your concerns. Your valuable feedback has significantly strengthened our work, and we are deeply grateful for the time and effort you have dedicated to reviewing our paper. Please let us know if there are any remaining questions or concerns we can address further. Thank you once again for your thoughtful comments!

---

> ### Author Response · Authors · 2024-12-01
>
> Dear Reviewer SHVx, we’ve carefully addressed all your insightful comments and incorporated all your suggestions in the revised manuscript.
>
> As the discussion period is nearing its end, I wanted to ensure you’ve had the opportunity to review our latest responses and updated paper. Please let us know if there are any remaining questions or concerns we can address further.
>
> Thank you once again for spending a significant amount of time on our submission and giving lots of valuable and insightful suggestions, which made our paper better!

---

### Official Review · Reviewer_wyfy · 2024-11-04

**Soundness:** 3
**Presentation:** 3
**Contribution:** 3
**Rating:** 6
**Confidence:** 4

**Summary:**

This paper addresses the multi-modal open-set TTA setting, a challenging and less-explored area within TTA research. It aims to adapt a source pre-trained multi-modal model to an unseen target domain containing samples from unknown classes. They propose an Adaptive Entropy-aware Optimization (AEO) framework, which encourages the adapted model to amplify entropy difference between known and unknown samples. Specifially, UAE dynamically determines whether to minimize or maximize the entropy of each sample through a sample-wise weighting scheme, while AMP considers modalities to encourage diverse predictions for unknown samples and consistent predictions for known samples. The AEO framework shows its effectiveness on the proposed MM-OSTTA benchmark across various modalities.

**Strengths:**

- The paper addresses the MM-OSTTA setting for the first time, which is a challenging but practical scenario in real-world applications.
- The paper is well written, making it easy to follow the introduced setting and proposed algorithms.
- The paper provides extensive experiments across two tasks and five modalities, showing the effectiveness and versatility of the proposed method.

**Weaknesses:**

- The observation in Sec. 3.1 is less impressive in the open-set settings. Previous works [1-2] on the open-set settings have already discovered and leveraged the entropy difference between known and unknown samples.

- In lines 171-174, the authors argue that some selective entropy minimization methods struggle to increase the entropy difference. But, the evidence supporting this claim is insufficient; Figure 2 compares only simple baselines (source, tent), missing another promising baseline such as EATA [3].

- EATA [3] also takes a sample-wise adaptive weighting scheme based on entropy values and achieves competitive performance. Such methods (e.g, EATA, SAR) seem to be able to increase the entropy difference between known and unknown samples, at least more effectively than TENT. What advantages does the proposed method offer over such selective entropy minimization methods?

- In Sec. 3.3, it is straightforward to enforce consistent predictions across modalities for known samples. However, for unknown samples, what is the benefit of maximizing prediction discrepancy across modalities? The equation (6) seems to penalize (i.g., maximize the prediction discrepancy) samples with consistently high entropy (likely unknown) across modalities, although they are easily classified as unknown. A more in-depth explanation of the adaptive modality prediction discrepancy loss in equation (6) would be valuable.

- One of the key purposes of TTA is to adapt the pre-trained model to make better predictions on unseen test data. Although the proposed AEO significantly improves unknown class detection (AUROC, FPR95), its accuracy gains are limited compared to the baselines. Also, the proposed framework primarily focuses on unknown class detection, showing limited capability in model adaptation itself.


[1] Safaei et al., Entropic open-set active learning. AAAI'24. \
[2] Gao et al., Unified Entropy Optimization for Open-Set Test-Time Adaptation. CVPR'24. \
[3] Niu et al., Efficient test-time model adaptation without forgetting. ICML'22.

**Questions:**

- Please refer to the weaknesses part.
- In equation (3), (5) and (6), how is the entropy used for the adaptive weight W_ada calculated? Is it based on predictions from each individual modality or from the aggregated output?
- For the corruption robustness experiments, there are six types of corruption for both videos and audios, resulting in 36 possible combinations (6*6) of corrupted test sets. Why are only six distinct corruption shifts used in the experiments?
- In Table 7, what happens if the diversity loss in equation (8) is excluded?

**Details Of Ethics Concerns:**

- There is no potential violation of the CoE.

---

> ### Author Response · Authors · 2024-11-18
> **The Response to Reviewer wyfy (Part 1)**
>
> Thanks for your insightful reviews, and we appreciate your valuable suggestions! We provide the responses to your questions as follows:
>
> >**Q1**: The observation in Sec. 3.1 is less impressive in the open-set settings. Previous works [1-2] on the open-set settings have already discovered and leveraged the entropy difference between known and unknown samples.
>
> **A1**: Thank you for your comment. It is indeed true that [1-2] utilized the entropy difference between known and unknown samples in open-set settings. However, our work goes beyond merely leveraging this difference. In Section 3.1, we provide **a deeper and more comprehensive analysis** of this phenomenon. Specifically, we investigate the relationship between entropy differences and the performance of MM-OSTTA across various baselines, identifying potential failure modes that previous works did not address. Building on these insights, we propose the Adaptive Entropy-aware Optimization (AEO) framework, specifically designed for the MM-OSTTA setup to amplify the entropy difference between known and unknown samples dynamically during online adaptation.
> ___
>
> >**Q2**: In lines 171-174, the authors argue that some selective entropy minimization methods struggle to increase the entropy difference. But, the evidence supporting this claim is insufficient; Figure 2 compares only simple baselines (source, tent), missing another promising baseline such as EATA [3].
>
> **A2**: Thank you for your insightful suggestion! We have updated Figure 2 to include another promising baseline SAR (Niu et al., ICLR 2023), which is a more recent approach compared to EATA (Niu et al., ICML 2022). As shown in Figure 2, while SAR outperforms Tent, it still fails to surpass the Source baseline in the multimodal open-set TTA setting. This observation highlights its limitation in effectively increasing the entropy difference between known and unknown samples, further validating our claim.
> ___
>
> >**Q3**: EATA [3] also takes a sample-wise adaptive weighting scheme based on entropy values and achieves competitive performance. Such methods (e.g, EATA, SAR) seem to be able to increase the entropy difference between known and unknown samples, at least more effectively than TENT. What advantages does the proposed method offer over such selective entropy minimization methods?
>
> **A3**: Thank you for pointing this out! Both SAR and EATA select reliable samples according to their entropy values for adaptation, excluding high-entropy samples from adaptation and assigning higher weights to test samples with lower prediction uncertainties. While SAR achieves better performance than Tent, it faces two key limitations in the multimodal open-set TTA setting:
>
> **1. Limited handling of unknown classes.**
> SAR focuses solely on minimizing the entropy of samples with lower prediction uncertainties (likely belonging to known classes) while leaving the high-entropy samples (potentially unknown classes) unaddressed. This selective approach restricts its ability to amplify the entropy difference between known and unknown samples. In contrast, our proposed Adaptive Entropy-aware Optimization (AEO) dynamically optimizes the entropy of both known and unknown classes, resulting in a significantly larger entropy difference, as demonstrated in Figure 2.
>
> **2. Sensitivity to overlapping score distributions.**
> In the challenging open-set setting, the initial score distributions of known and unknown samples are often closely aligned and difficult to distinguish, as shown in Figure 6 (a). In such cases, SAR could potentially minimize the entropy of unknown samples incorrectly, leading to degraded performance that may even fall below the Source baseline. AEO addresses this issue by progressively increasing the entropy difference between known and unknown samples, mitigating the risk of negative adaptation and ensuring more reliable performance.
>
> Additionally, we introduce **Adaptive Modality Prediction Discrepancy Optimization**, which exploits cross-modal interactions to further enhance the separation between known and unknown classes. These complementary approaches strengthen our framework's ability to handle open-set challenges more effectively than selective entropy minimization methods.

---

> ### Author Response · Authors · 2024-11-18
> **The Response to Reviewer wyfy (Part 2)**
>
> >**Q4**: In Sec. 3.3, it is straightforward to enforce consistent predictions across modalities for known samples. However, for unknown samples, what is the benefit of maximizing prediction discrepancy across modalities? The equation (6) seems to penalize (i.g., maximize the prediction discrepancy) samples with consistently high entropy (likely unknown) across modalities, although they are easily classified as unknown. A more in-depth explanation of the adaptive modality prediction discrepancy loss in equation (6) would be valuable.
>
> **A4**: Thank you for your thoughtful question! The Adaptive Modality Prediction Discrepancy Optimization (AMP) is designed to further enhance the entropy difference between known and unknown samples by leveraging the complementary information from multiple modalities. It achieves this by adaptively enforcing consistency or discrepancy in predictions across modalities based on whether the samples are likely to belong to known or unknown classes.
>
> For **known samples**, enforcing consistent predictions across modalities helps the model make confident and accurate final predictions, as each modality supports the others.
>
> For **unknown samples**, enforcing consistent predictions across modalities can be detrimental. If both modalities have predictions with high entropy on unknown samples, enforcing consistent predictions across them has minimal impact. However, when the model suffers from overconfidence issues and outputs high prediction certainty (low entropy) for unknown class samples (Figure 6 (a)), enforcing consistent predictions can make the model output final predictions with high certainty aligned with known classes, degrading the open-set performances. As shown in our ablation in Table 1, enforcing consistent predictions across modalities degrades the performances on all datasets.
>
> |                 | HAC | EPIC-Kitchens | Kinetics-100-C |
> |-----------------|-----|---------------|----------------|
> | Consistent Prediction |  47.61   |        55.12       |          58.91      |
> | AEO (ours)      |    **48.31** |       **56.18**        |      **60.07**          |
>
> *Table 1: Ablation on consistent predictions across modalities to the final performances. The H-score is reported.*
>
> We also investigate the influence of each component in AMP, including $L_{AdaEnt*}$ and $L_{AdaDis}$, The $L_{AdaEnt*}$ term adaptively maximizes or minimizes the prediction entropy of each modality. $L_{AdaDis}$, dynamically adjusts the prediction discrepancy between modalities based on whether the sample is likely known or unknown. As shown in Table 2, both components contribute to performance improvements, and their combination yields the best results. By adaptively balancing consistency and discrepancy across modalities, AMP fully exploits modality interactions to address the unique challenges of the multimodal open-set TTA setting.
>
> | $L_{AdaDis}$ | $L_{AdaEnt*}$ | Acc↑  | FPR95↓ | AUROC↑ | H-score↑        |
> |------------------------|-------------------------|-------|--------|--------|------------------|
> | $\checkmark$           |                         | 59.49 | 67.48  | 72.39  | 47.60     |
> |                        | $\checkmark$            | 59.85 | 67.41  | 72.36  | 47.74      |
> | $\checkmark$           | $\checkmark$            | 59.53 | 66.75  | 72.50  | **48.31**  |
>
> *Table 2: Ablation on each component in AMP.*

---

> ### Author Response · Authors · 2024-11-18
> **The Response to Reviewer wyfy (Part 3)**
>
> >**Q5**: One of the key purposes of TTA is to adapt the pre-trained model to make better predictions on unseen test data. Although the proposed AEO significantly improves unknown class detection (AUROC, FPR95), its accuracy gains are limited compared to the baselines. Also, the proposed framework primarily focuses on unknown class detection, showing limited capability in model adaptation itself.
>
> **A5**: Thanks for your insightful comment! In multimodal open-set TTA, model adaptation serves a dual purpose: improving the accuracy of **known samples** (closed-set adaptation) and effectively detecting **unknown samples** (open-set adaptation), and there is typically a trade-off between them. Both objectives are critical, particularly for safety-critical applications like autonomous driving and fraud detection, where robust unknown sample detection is vital for reliability.
>
> For the closed-set adaptation ability to known samples, we observe a common phenomenon in existing multimodal TTA work [a], where the improvement in accuracy is low compared to the Source baseline. In [a], they report improvements of only 2.6% on Kinetics50-C and 0.9% on VGGSound-C compared to the Source baseline. As a comparison, in our MM-OSTTA setting, our method improves the accuracy of 0.92% on EPIC-Kitchens dataset, 3.17% on HAC dataset, and 10.92% on Kinetics-100-C compared to the Source baseline. Compared to other SOTA baselines, our AEO is consistently among the top performers, with differences in accuracy being less than 1% in most cases compared to the best baseline.
>
> Furthermore, we observed that Tent, which minimizes the entropy of all samples indiscriminately (whether known or unknown), also improves the closed-set accuracy of 3.0% on HAC dataset, and 8.89% on Kinetics-100-C compared to the Source baseline (with terrible open-set adaptation performances on AUROC and FPR95). This observation suggests that adapting improperly to unknown samples can sometimes improve closed-set accuracy, while substantially explaining why other baseline methods also achieve competitive results on closed-set accuracy although with poor open-set performances.
>
> For open-set adaptation ability to unknown samples, our **AEO demonstrates significant superiority** over other baselines, improving the FPR95 of 30.47% on EPIC-Kitchens dataset, 11.49% on HAC dataset, and 22.57% on Kinetics-100-C compared with the Source baseline, and 17.42%, 6.08%, 2.11% respectively compared with the best baseline.
>
> Our AEO uniquely balances both closed-set and open-set adaptation, achieving strong performance across both objectives. This capability is **crucial for real-world safety-critical applications**, where both accurate predictions on known classes and reliable identification of unknown samples are essential for system robustness and reliability.
>
> We hope this clarifies the significant advantages of AEO and its ability to meet the dual objectives of multimodal open-set TTA. Thank you for helping us highlight these key aspects of our work.
>
> [a] Yang et al, Test-time Adaptation against Multi-modal Reliability Bias, in ICLR 2024
> ___
> >**Q6**: In equation (3), (5) and (6), how is the entropy used for the adaptive weight W\_ada calculated? Is it based on predictions from each individual modality or from the aggregated output?
>
> **A6**: Sorry for the confusion. In eq. (4-6), We use the same $W_{ada}$ calculated in eq. (3). It is based on predictions from the aggregated output $\hat{p}$.

---

> ### Author Response · Authors · 2024-11-18
> **The Response to Reviewer wyfy (Part 4)**
>
> >**Q7**: For the corruption robustness experiments, there are six types of corruption for both videos and audios, resulting in 36 possible combinations (6*6) of corrupted test sets. Why are only six distinct corruption shifts used in the experiments?
>
> **A7**: Thanks for your insightful comment! Evaluating all 36 possible combinations is indeed an interesting direction. However, displaying results for all combinations in a single table would be redundant and difficult to interpret. To maintain clarity and conciseness, we initially opted to evaluate six representative combinations selected randomly.
>
> In response to your suggestion, we have conducted additional experiments on all 36 combinations of corruptions. As shown in Figure 11 and Table 13 in the revised paper (and summarized in Table 3 below), **our AEO achieves the best performance on 31 out of 36 combinations** and obtains the highest average H-score, further demonstrating its robustness across diverse corruption scenarios.
>
> |                         | Source | Tent  | SAR   | OSTTA | UniEnt | READ  | AEO (ours) |
> |-------------------------|--------|-------|-------|-------|--------|-------|------------|
> | **H-score**             | 37.30  | 16.90 | 16.11 | 28.14 | 45.38  | 59.11 | **60.56**  |
>
> *Table 3: Average H-score on all 36 possible combinations of corruptions.*
> ___
> >**Q8**: In Table 7, what happens if the diversity loss in equation (8) is excluded?
>
> **A8**: Thanks for your insightful question! As shown in Table 4 below (detailed results are in Table 16 in the revised paper), without $L_{Div}$, the performance on the EPIC-Kitchens dataset is close to one with $L_{Div}$, but the performances drop significantly for HAC and Kinetics-100-C datasets, verifying the important of $L_{Div}$ to ensure diversity in predictions.
>
> |                 | HAC | EPIC-Kitchens | Kinetics-100-C |
> |-----------------|-----|---------------|----------------|
> | w/o $L_{Div}$ |  46.62   |         **56.40**       |         50.15      |
> | w/ $L_{Div}$     |    **48.31** |      56.18        |      **60.07**          |
>
> *Table 4: Ablation on $L_{Div}$ to the final performances. The H-score is reported.*

---

> ### Author Response · Authors · 2024-11-24
>
> Dear Reviewer wyfy, we’ve carefully addressed all your insightful comments and incorporated your suggestions in the revised manuscript. Your feedback has been invaluable in improving the quality of our work, and we greatly appreciate your thoughtful review.
>
> As the discussion period is nearing its end, I wanted to ensure you’ve had the opportunity to review our responses and updated paper. Please let us know if there are any remaining questions or concerns we can address further.
>
> Thank you once again for your time and consideration.

---

> > ### Comment · Reviewer_wyfy · 2024-11-25
> >
> > Thank you for your detailed responses.
> >
> > After reviewing your responses, several concerns have been resolved, but some still remain.
> >
> > The major remaining concerns are as follows:
> > 1) Regarding performance improvement in terms of accuracy, the reviewer agrees that the multimodal open-set TTA serves dual purposes: closed-set adaptation and unknown sample detection. However, the reviewer's original concern was that AEO's algorithmic design and motivation, as well as its performance are mostly focused on the latter purpose. As the author's argument, there is typically a trade-off between the two purposes, but the proposed AEO seems to have its advantages at the cost of some loss in accuracy compared to the baselines.
> >
> > 2) Regarding the diversity loss in equation (8), the authors have provided the ablation study on it, but the performance drop in the absence of the diversity loss could hinder the solidity of the proposed method. Since the ablation results only include the H-score, the reviewer still has questions about which metrics are mainly affected by the exclusion of the diversity loss.
> >
> > Once again, the reviewer appreciates the authors for their detailed responses.
> >
> > Sincerely, \
> > Reviewer wyfy

---

> > > ### Author Response · Authors · 2024-11-25
> > >
> > > Dear Reviewer wyfy,
> > >
> > > We are delighted to hear that several of your concerns have been resolved, and we sincerely appreciate your continued engagement and insightful feedback. Below, we address your remaining concerns in detail.
> > >
> > > >**Regarding performance improvement in terms of accuracy**
> > >
> > > Thank you for raising this important concern. While our AEO framework is indeed primarily designed to enhance unknown sample detection, it **does not compromise accuracy (Acc)** for closed-set adaptation. On the contrary, our results show that AEO achieves competitive accuracy across all datasets.
> > >
> > > To clarify, the assertion that AEO sacrifices Acc for its advantages in unknown sample detection is **not supported by our findings**. For example, while READ excels on EPIC-Kitchens and Kinetics-100-C, it performs poorly on the HAC dataset. Conversely, SAR performs well on HAC dataset but poorly on EPIC-Kitchens and Kinetics-100-C datasets. Our AEO, however, consistently performs well across all datasets. To provide further clarity, we have **computed the average Acc across all datasets** (EPIC-Kitchens, HAC (video+audio, video+flow, flow+audio, video+audio+flow), and Kinetics-100-C). As shown in Table 1 below, our **AEO achieves the highest overall performance in Acc**, demonstrating its robustness.
> > >
> > > || Source | Tent  | SAR   | OSTTA | UniEnt | READ  | AEO (ours) |
> > > |-------------------------|--------|-------|-------|-------|--------|-------|------------|
> > > | **Acc↑**             | 53.02| 54.97| 56.00 |54.65| 55.77 |55.77 | **56.30**  |
> > >
> > > *Table 1: Average accuracy (Acc) across all datasets.*
> > >
> > > Moreover, we have computed the average values of other metrics (FPR95, AUROC, and H-score) across all datasets. As shown in Table 2, **AEO achieves the best overall performance across all metrics**, with notable improvements in FPR95, AUROC, and H-score. While the improvement in Acc is relatively modest, it is significant that AEO balances both closed-set adaptation and unknown sample detection effectively.
> > >
> > > |  | Acc↑  | FPR95↓ | AUROC↑ | H-score↑        |
> > > |-------------------------|-------|--------|--------|------------------|
> > > |  Source|  53.02 | 77.58 | 61.86 | 35.78|
> > > |    Tent|54.97 | 88.84 | 55.40 | 22.68|
> > > |    SAR |56.00 | 85.36 | 60.94 | 26.71|
> > > |  OSTTA|54.65 | 86.40 | 59.24 | 26.23|
> > > |    UniEnt|55.77 | 71.51 | 70.22 | 41.62|
> > > |    READ | 55.77 | 72.14 | 69.36 | 42.81|
> > > |    AEO (ours) |**56.30 (+0.30)** | **61.93 (+9.58)** | **76.82 (+6.60)** | **50.82 (+8.01)**|
> > >
> > > *Table 2: Average value of different metrics on all datasets.*
> > >
> > > In summary, our work is the **first** to tackle the challenging and practical problem of Multimodal Open-set Test-time Adaptation. AEO does not trade Acc for improved unknown sample detection but instead **achieves a well-rounded performance across both closed-set and open-set adaptation objectives**.
> > >
> > >
> > >
> > > >**Regarding the diversity loss in equation (8)**
> > >
> > > The detailed results of all metrics are shown in Table 16 in our revised paper. For HAC dataset, FPR95 and AUROC are mainly affected by the exclusion of the diversity loss. For Kinetics-100-C, all three metrics are affected. Diversity loss is a common practice in TTA to promote prediction diversity and prevent model collapse. **To ensure fair comparisons, we included this loss term in all baselines as well.**
> > >
> > > We sincerely hope these updates and clarifications address your concerns. Your valuable feedback has significantly strengthened our work, and we are deeply grateful for the time and effort you have dedicated to reviewing our paper. Thank you once again for your thoughtful comments!

---

> > > ### Author Response · Authors · 2024-11-27
> > >
> > > Dear Reviewer wyfy,
> > >
> > > We deeply appreciate your thoughtful feedback, which has helped us strengthen our work. Based on your insightful comments, we have incorporated all additional ablations and analyses into the final version of the revised manuscript. Below is a summary of our updates:
> > >
> > > We have added Section **C.14.3 COMPARISON WITH OPEN-SET METHODS** to compare our work with existing open-set work [1] and [2].
> > >
> > > We have added Section **C.14.4 COMPARISON WITH SELECTIVE ENTROPY MINIMIZATION METHODS** to discuss the limitations of EATA [3] and SAR [4] and highlight the unique advantages of our AEO.
> > >
> > > We have added Section **C.14.2 COMPARISON WITH MULTIMODAL TTA METHODS and Tab. 25** to provide a more in-depth explanation of the adaptive modality prediction discrepancy loss.
> > >
> > > We have added Section **C.14.5 PERFORMANCE IMPROVEMENT IN ACCURACY and Tab. 26-27** to address your concerns regarding the Acc metric.
> > >
> > > We have added Section **C.2 EVALUATION RESULTS ON ALL 36 POSSIBLE COMBINATIONS OF CORRUPTIONS, Fig. 11 and Tab. 13** to include results on all 36 possible combinations of corruptions.
> > >
> > > We have added Section **C.5 INFLUENCE OF $L_{Div}$ TO THE PERFORMANCES and Tab. 16** to include detailed ablations on $L_{Div}$.
> > >
> > > [1] Safaei et al., Entropic open-set active learning. AAAI'24.
> > >
> > > [2] Gao et al., Unified Entropy Optimization for Open-Set Test-Time Adaptation. CVPR'24.
> > >
> > > [3] Niu et al., Efficient test-time model adaptation without forgetting. ICML'22.
> > >
> > > [4] Niu et al., Towards stable test-time adaptation in dynamic wild world. ICLR'23.
> > >
> > > We sincerely hope these updates and clarifications address your concerns. Your valuable feedback has significantly strengthened our work, and we are deeply grateful for the time and effort you have dedicated to reviewing our paper. Please let us know if there are any remaining questions or concerns we can address further. Thank you once again for your thoughtful comments!

---

> > > ### Author Response · Authors · 2024-11-30
> > > **Kind Reminder Regarding Our Response to Your Comments**
> > >
> > > Dear Reviewer wyfy,
> > >
> > > We hope this message finds you well. We would like to kindly remind you of our response to your concerns about performance improvement in accuracy. We sincerely hope the additional analyses and results we provided clarify and resolve your concerns.
> > >
> > > As the discussion period is nearing its end, we greatly value your insights and would deeply appreciate it if you could revisit our response and consider reevaluating our paper. Your feedback has been immensely helpful in strengthening our work, and we hope to have addressed your points thoroughly.
> > >
> > > Thank you again for your time and thoughtful comments. Please let us know if there are any remaining questions or aspects that require further clarification.

---

> > > ### Author Response · Authors · 2024-12-01
> > >
> > > Dear Reviewer wyfy, we’ve carefully addressed all your insightful comments and incorporated all your suggestions in the revised manuscript.
> > >
> > > As the discussion period is nearing its end, I wanted to ensure you’ve had the opportunity to review our latest responses and updated paper. Please let us know if there are any remaining questions or concerns we can address further. We would deeply appreciate it if you could revisit our response and consider reevaluating our paper.
> > >
> > > Thank you once again for spending a significant amount of time on our submission and giving lots of valuable and insightful suggestions, which made our paper better!

---

> ### Comment · Reviewer_wyfy · 2024-12-02
> **Final decision**
>
> Thank you for your detailed responses and clarifications.
>
> After reviewing all the responses and the revised paper again, I find that my original concerns and questions are mostly addressed during the rebuttal period.
>
> Therefore, I have decided to raise my score from 5 to 6.
>
> To further improve the paper, I recommend clarifying some important aspects, especially regarding Q3, Q4, and Q5 in the next version of the paper.
>
> Once again, I appreciate the authors' comprehensive responses.

---

> > ### Author Response · Authors · 2024-12-02
> > **Thanks for raising the score!**
> >
> > Dear Reviewer wyfy,
> >
> > We are glad to hear that we have addressed most of your concerns and that you raised your score to 6! We have added clarification on Q3, Q4, and Q5 in the Appendix and will integrate them into the main paper for the final version. Thanks again for spending a significant amount of time on our submission and giving lots of valuable and insightful suggestions, which make our paper even stronger!

---

### Author Response · Authors · 2024-11-18
**Highlight on Main Contributions and Summary of Revisions**

We sincerely thank all the reviewers for their thoughtful and constructive feedback. We have carefully read through them and provided global and individual responses, respectively. In global responses here, we highlight our main contributions and provide a detailed summary of the revisions made to the paper.

>**Main Contributions**

**Introducing the First Work on MM-OSTTA.** We tackle the challenging and practical problem of Multimodal Open-set Test-time Adaptation (MM-OSTTA) for the first time.

**Proposing a Novel Adaptive Entropy-aware Optimization (AEO) Framework.**  Our proposed AEO framework is designed to amplify the entropy difference between known and unknown samples, based on the observation that the entropy difference strongly correlates with MM-OSTTA performance. AEO integrates two novel components: Unknown-aware Adaptive Entropy Optimization (UAE) and Adaptive Modality Prediction Discrepancy Optimization (AMP). UAE adaptively weights and optimizes the entropy of each sample based on its prediction uncertainty. AMP dynamically adjusts prediction discrepancies between modalities to fully exploit modality interactions and further enhance entropy separation for known and unknown samples.

**Comprehensive Benchmarking Across Tasks, Datasets, and Modalities.**
We benchmark our approach on **four datasets** (EPIC-Kitchens, Human-Animal-Cartoon, Kinetics-100-C, and nuScenes datasets) spanning **two tasks** (action recognition and 3D semantic segmentation) and **five modalities** (video, audio, optical flow, LiDAR point cloud, RGB image). In contrast, most prior TTA works focused solely on single-modality classification tasks.

**Robustness Demonstrated Through Extensive Settings.**
We go beyond the traditional MM-OSTTA setup and validate the robustness of our method under challenging scenarios, including **long-term** MM-OSTTA, **continual** MM-OSTTA, and MM-OSTTA with **mixed distribution shifts**.

**Comprehensive Ablation Studies and Analyses.** We provide extensive ablation studies to analyze the impact of each proposed module and each hyperparameter. We further explore the influence of different architectures, pre-trained models, score functions, severity levels of corruptions, combination of corruptions, and the ratio of unknown samples, offering a thorough understanding of the proposed approach.

>**Changes Made in the Revised Paper**

We discussed the limitation of existing work in the introduction from Line 072-075.

We updated Figure 2 to include an additional baseline SAR [a].

We added **sections C.1-C.8, C.14, Tables 12-18, 25-27, and Figures 11-14 in the Appendix** to provide additional experiments and analyses as requested by the reviewers. We believe these updates further strengthen the contributions and rigor of our work, addressing all reviewer concerns thoroughly and effectively. **We highlight all changes with blue color in the revised paper.**

[a] Niu et al. Towards Stable Test-Time Adaptation in Dynamic Wild World. ICLR 2023

---

### Meta-Review · Area_Chair_BuBw · 2024-12-16

**Metareview:**

All the 3 reviewers provide positive ratings after rebuttal, with 2 upgraded scores. Initially, the reviewers had concerns about some technical clarity/novelty, e.g., using the entropy and performance improvement. In the post-rebuttal discussion period, all the reviewers are satisfactory with the authors' comments and revised paper. After taking a close look at the paper, rebuttal, and discussions, the AC agrees with reviewers' feedback of the proposed method being effective to solve a challenging and practical task for multimodal open-set test-time adaptation. Therefore, the AC recommends the acceptance rating based on the discussion contents and revised paper.

**Additional Comments On Reviewer Discussion:**

Initially, all the reviewers had concerns about the limited performance improvements. In post-rebuttal discussion, the authors provided more results and explanations across both closed-set and open-set settings, in which the reviewers are satisfactory with the new results being more convincing. Moreover, the reviewer uXos requested more comparisons with the entropy-based relevant work, and was happy with the authors' response. The AC considers all the feedback and agrees with reviewers' assessment.

---

### Decision · Program_Chairs · 2025-01-22

Accept (Poster)